# CDK9 activity switch associated with AFF1 and HEXIM1 controls differentiation initiation from epidermal progenitors

Sarah M. Lloyd [1,2], Daniel B. Leon[1], Mari O. Brady [1], Deborah Rodriguez [1], Madison P. McReynolds[1], Junghun Kweon[1], Amy E. Neely[1], Laura A. Blumensaadt[1], Patric J. Ho[1] & Xiaomin Bao [1,2,3,4] ✉

Progenitors in epithelial tissues, such as human skin epidermis, continuously make fate decisions between self-renewal and differentiation. Here we show that the Super Elongation Complex (SEC) controls progenitor fate decisions by directly suppressing a group of "rapid response" genes, which feature high enrichment of paused Pol II in the progenitor state and robust Pol II elongation in differentiation. SEC's repressive role is dependent on the AFF1 scaffold, but not AFF4. In the progenitor state, AFF1-SEC associates with the HEXIM1-containing inactive CDK9 to suppress these rapid-response genes. A key rapid-response SEC target is ATF3, which promotes the upregulation of differentiation-activating transcription factors (GRHL3, OVOL1, PRDM1, ZNF750) to advance terminal differentiation. SEC peptidomimetic inhibitors or PKC signaling activates CDK9 and rapidly induces these transcription factors within hours in keratinocytes. Thus, our data suggest that the activity switch of SEC-associated CDK9 underlies the initial processes bifurcating progenitor fates between self-renewal and differentiation.

Self-renewing somatic tissues are subject to constant turnover. In this process, a subset of progenitors proliferates to sustain their population, while another subset progressively activates terminal differentiation to fulfill specialized tissue functions. The gene regulatory mechanisms governing these two distinct progenitor fate choices, especially the early events that initiate the terminal differentiation process, remain incompletely understood.

Skin epidermis, composed of 90% keratinocytes, is a type of highly accessible self-renewing somatic tissue. Multiple signaling pathways including calcium as well as Protein Kinase C (PKC) signaling are closely linked to the activation of keratinocyte differentiation[1,2]. Recent findings have further identified a group of keratinocyte differentiation activators including the transcription factors ZNF750, OVOL1, GRHL3, and PRDM1[3–7]. These activators are strongly upregulated during the early differentiation stage and subsequently activate

target genes associated with epidermal-barrier function. It remains unclear how these differentiation activators are upregulated during early differentiation and what enforces their repression in the progenitors.

RNA Polymerase II (Pol II) pause release has emerged as an evolutionarily conserved mechanism for rapidly activating gene expression. In this context, Pol II accumulates and pauses near a subset of promoters waiting for specific signaling to switch it towards productive RNA synthesis along the gene bodies. Pol II pause release was initially characterized by *Drosophila* heat shock genes[8]. This phenomenon has been subsequently observed in several developmental processes including embryonic development and hematopoiesis[9]. The molecular events enforcing Pol II pausing and elongation in different developmental processes including the early events of progenitor fate choices require further investigation.

[1]Department of Molecular Biosciences, Northwestern University, Evanston, IL 60208, USA. [2]Simpson Querrey Institute for Epigenetics, Northwestern University, Chicago, IL 60611, USA. [3]Robert H. Lurie Comprehensive Cancer Center, Northwestern University, Chicago, IL 60611, USA. [4]Department of Dermatology, Northwestern University, Chicago, IL 60611, USA. ✉e-mail: xiaomin.bao@northwestern.edu

The CDK9 kinase plays a central role in regulating Pol II pause release[10]. CDK9 and a Cyclin T regulatory subunit are collectively known as "positive transcription elongation factor b" (PTEFb). A number of substrates have been established for CDK9, including the Pol II C-terminal domain (CTD) as well as the negative elongation factor (NELF). Phosphorylation of Pol II CTD switches Pol II's association from the transcriptional initiation apparatus to the elongation and splicing machinery[11]. Phosphorylation of NELF disrupts its association with Pol II, releasing Pol II into productive elongation[12].

The kinase activity of CDK9 is strictly regulated through incorporation into distinct complexes to prevent promiscuous gene induction. The inactive form of CDK9 is tethered by the 7SK snRNP complex that includes HEXIM1 as an essential component[13–15]. In particular, HEXIM1 directly interacts with the activation segment of CDK9 to inhibit its kinase activity[16]. The active form of CDK9 is commonly associated with the Super Elongation Complex (SEC)[10,17] or the bromodomain protein 4 (BRD4)[10]. How CDK9 activity is modulated in the context of progenitor function in self-renewing somatic tissues remains under-characterized.

Here we show that the AFF1-containing SEC associates with the HEXIM1-containing inactive CDK9 to directly suppress a group of rapid-response genes in the progenitor state keratinocytes. These rapid-response genes controlled by the SEC are characterized by high enrichment of paused Pol II in the progenitor state and robust Pol II elongation in differentiation. Treating keratinocytes with SEC peptidomimetic inhibitor (KL1 or KL2), which mimics the "LFAEP" sequence in the N-terminal region of AFF4 (and "LFGEP" in AFF1)[18] rapidly derepresses the target genes. Among these rapid-response targets directly repressed by AFF1-SEC, we identified ATF3 as a key initiator for the differentiation process. ATF3 upregulation is sufficient to subsequently promote the expression of several differentiation-driving transcription factors including ZNF750, OVOL1, GRHL3, and PRDM1. We further show that the CDK9 activity switch, associated with AFF1 and HEXIM1, mediates the early events of PKC signaling in initiating keratinocyte differentiation. Mechanistically, we found that PKC signaling or SEC peptidomimetic inhibitor disrupts CDK9's association with HEXIM1 in keratinocytes, switching CDK9 activity from "off" to "on". This activation of CDK9 is required to upregulate both the rapid-response genes and differentiation-driving transcription factors. Taken together, our data demonstrate that AFF1's association with inactive CDK9 at the promoter-proximal regions of early-response genes is essential for progenitor maintenance; rapid activation of these genes, through CDK9 activity switch, promotes the expression of differentiation-activating transcription factors (TFs) to further advance the terminal differentiation process.

## Results

### High Pol II binding at differentiation or KL-induced genes

To determine the association between Pol II dynamics and gene expression during epidermal differentiation, we generated Pol II ChIP-seq data in undifferentiated (progenitor state) and differentiated (calcium-induced differentiation, day 4) primary human keratinocytes. These keratinocytes were differentiated by seeding at 100% confluency and adding 1.2 mM $CaCl_2$ to the media. We compared these Pol II ChIP data with our recently published RNA-seq data profiling gene expression in these two states[5] (Fig. 1a, Supplementary Data 1). These included 3593 upregulated genes and 2692 downregulated genes (fold change ≥2, $p < 0.05$, two-tailed, Wald test). To explore distinct mechanisms underlying gene upregulation in the differentiation process, we applied k-means clustering based on Pol II ChIP enrichment in both undifferentiation and differentiation conditions. This unbiased approach unveiled two groups of upregulated genes, "cluster I" and "cluster II". Genes associated with "cluster I" feature high Pol II enrichment in the progenitor state and a signature of pause release in differentiation. These genes include the dual-specificity phosphatase

DUSP1, which has been previously observed as a rapidly upregulated gene in keratinocyte differentiation induced by suspension[19] (Fig. 1b, c). Genes associated with "cluster II", such as the epidermal differentiation-activating transcription factor, GRHL3, had much lower overall Pol II enrichment as compared to "cluster I" (Fig. 1d, e). Downregulated genes ("cluster III"), such as the proliferation regulator AURKB, showed a general decrease in Pol II occupancy during differentiation (Fig. 1f, g). In particular, "cluster I" genes showed the highest Pol II enrichment in TSS's among all three clusters, in both undifferentiated and differentiation states (****$p < 0.001$, unpaired $t$ test, Fig. 1h, i). Consistent with expression changes, both "cluster I" and "cluster II" genes had increased Pol II enrichment in the gene bodies in the differentiated state, while "cluster III" genes had decreased Pol II enrichment ($p < 0.001$, unpaired $t$ test, Fig. 1j). We also calculated and compared pausing indices between "clusters I" and "cluster II" (Supplementary Fig. 1a), to evaluate Pol II pause release as a mechanism for controlling the expression of these genes. Genes from both "cluster I" and "cluster II" showed a decrease in pausing indices in differentiation ($p < 0.001$, unpaired $t$ test, Supplementary Fig. 1b). Taken together, "cluster I" stood out from our integrated analyses of Pol II ChIP-seq and RNA-seq data as a unique group of upregulated genes in differentiation, featuring highly enriched paused Pol II in the undifferentiated state and robust Pol II elongation in differentiation.

To explore how Pol II dynamics are controlled between the undifferentiated and differentiated states, we investigated the role of the SEC in progenitor state keratinocytes. We leveraged the peptidomimetic inhibitors of the SEC, KL1, and KL2, which were designed based on the N-terminal peptide sequences of the SEC scaffolding protein AFF1 or AFF4[18]. We found that 3-hour KL (KL1 or KL2) treatment significantly altered the expression of 199 genes (fold change ≥2, $p < 0.05$, two-tailed, Wald test, Supplementary Data 2). These genes were primarily activated, rather than repressed, by KL (Fig. 1k). Gene Ontology (GO) term analysis showed that the upregulated genes, but not the downregulated genes, were highly enriched with the molecular functions related to "transcription factor activity" and "sequence-specific DNA binding" (Fig. 1l). We then asked if these genes, rapidly altered by KL in 3 hours, were associated with a specific cluster of Pol II enrichment. We found that these 199 genes are most highly correlated with "cluster I" (Fig. 1m), which consisted of genes upregulated in differentiation, strongly Pol II enriched, and with a signature of pause release in differentiation. Furthermore, the upregulated genes by 3-hour KL treatment had substantially stronger enrichment of Pol II in promoters in epidermal progenitors, as compared to the downregulated genes (Fig. 1n, o). Thus, treating the progenitor state keratinocytes with KL rapidly activates a subset of genes in "cluster I".

### KL induces differentiation and impairs proliferation

Given the rapid induction of genes related to transcription regulation with 3 hours of KL treatment, we asked if a longer-term KL treatment would cast a stronger impact on keratinocyte growth and gene expression. After 24 hours of KL treatment, keratinocytes showed drastically reduced proliferation as compared to DMSO control (Fig. 2a). Ki67 immunofluorescent labeling was strongly reduced with 24 hour of KL treatment (Fig. 2b, c). MYC and DNMT1 protein levels were also reduced (Supplementary Fig. 2a, b). On the other hand, the cell-cycle inhibitor p21 which is associated with keratinocyte differentiation[20,21] was strongly upregulated in KL-treated keratinocytes after 24 hours (Supplementary Fig. 2c, d). These data demonstrate that longer-term KL treatment is sufficient to impair keratinocyte proliferation, and this may also induce differentiation.

We then performed RNA-seq analysis comparing keratinocytes treated with KL versus DMSO control for 24 hours and identified a total of 1917 significantly altered genes, including 972 upregulated genes and 945 downregulated genes (fold change ≥2, $p < 0.05$, two-tailed, Wald test, Supplementary Data 3). 67% of these differentially

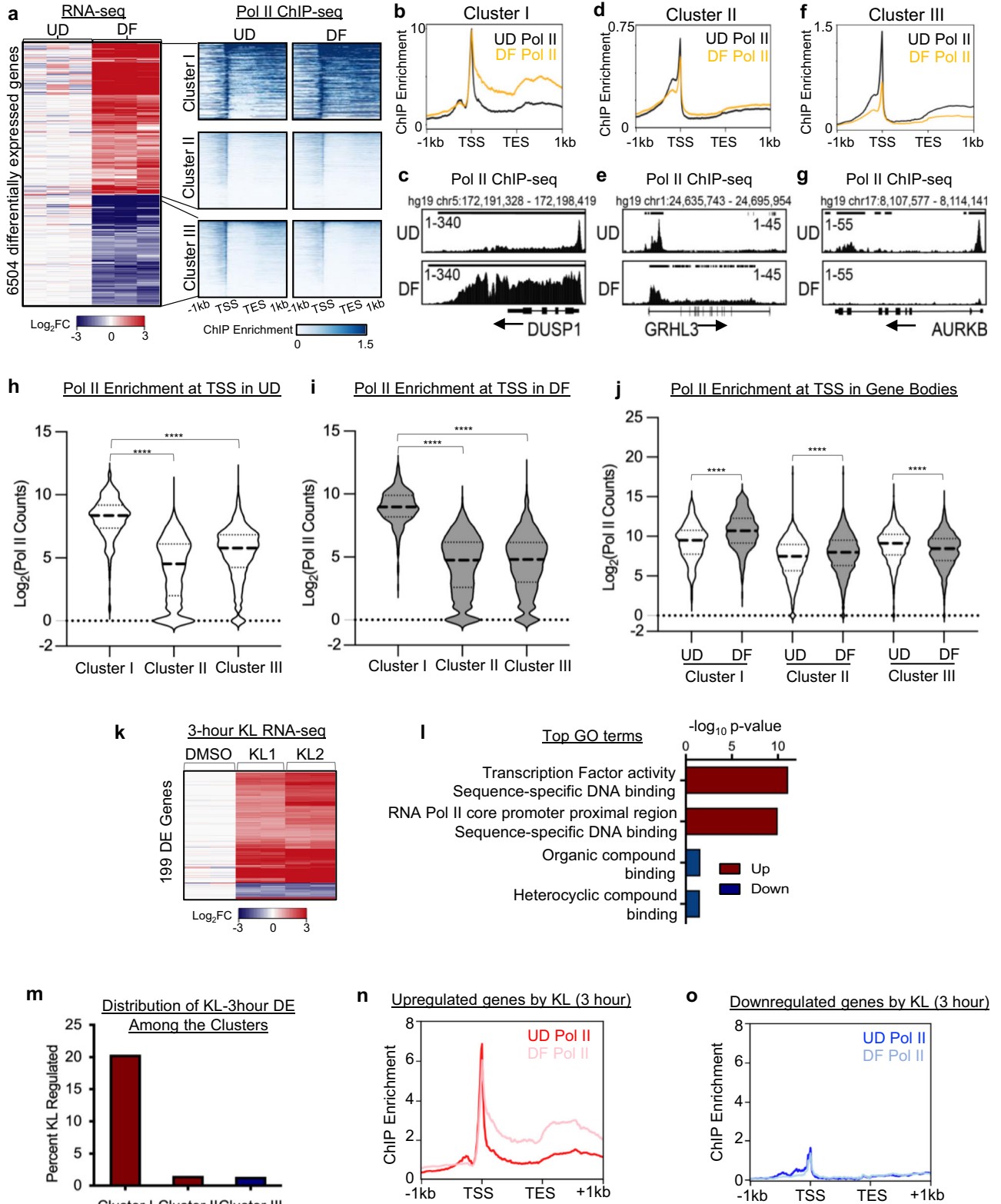

expressed genes overlapped with the calcium-induced keratinocyte differentiation signature (Fig. 2d). Notably, these upregulated genes included key differentiation-activating TFs, including GRHL3, OVOL1, PRDM1, and ZNF750. The upregulated genes demonstrated high enrichment of GO terms related to "establishment of skin barrier" and "epidermis development". The downregulated genes were related to "DNA replication" and "cell division" (Fig. 2e). Using qRT-PCR, we further confirmed and compared the expression kinetics of the

differentiation-activating TFs (GRHL3, OVOL1, PRDM1, ZNF750) and the proliferation markers (MKI67, AURKB, FOXM1) at 3 hours and 24 hours of KL treatment. The induction of these differentiation-activating TFs was already initiated at 3 hours of KL treatment, and the expression of these TFs continued to rise at the 24-hour time point (Fig. 2f, g). The proliferation markers were drastically downregulated only at the 24-hour, but not the 3-hour time point of KL treatment (Fig. 2h, i). The genes differentially expressed with 24-hour KL

**Fig. 1 | KL treatment rapidly activates a subset of differentiation-upregulated genes with high Pol II enrichment. a** Heatmaps showing the comparison of RNA-seq and Pol II ChIP-seq data of 6504 genes differentially expressed in undifferentiated (UD) and differentiated (DF) keratinocytes. K-means clustering was applied to Pol II ChIP-seq data corresponding to upregulated genes to create clusters I and II. Cluster III consists of all genes downregulated in keratinocyte differentiation. **b–g** Average profile plots of Pol II ChIP-seq data and representative examples for all three clusters. **h, i** Violin plots comparing Pol II ChIP-seq enrichment in the promoters of genes in cluster I, cluster II and cluster III, in undifferentiated (UD) and differentiated (DF) keratinocytes (****$P < 0.0001$, two-tailed, unpaired $t$ test) **j** Violin plot comparing total Pol II enrichment in UD gene bodies to DF gene body enrichment (****$P < 0.0001$, two-tailed, unpaired $t$ test). **k** RNA-seq heatmap showing the relative expression of the 199 genes differentially expressed (fold change ≥2, $P < 0.05$, two-tailed, Wald test) with 3 hours of KL1 or KL2 treatment. **l** Bar graph showing the top Gene Ontology (GO) terms of the upregulated and downregulated genes in keratinocytes treated with KL or DMSO for 3 hours (two-tailed, Fisher's exact test). **m** Percent of genes in clusters I, II, and III (**a**) that are differentially expressed with 3-hour KL treatment. **n, o** Average profile plots comparing Pol II ChIP-seq enrichment in the upregulated or downregulated genes after 3 hours of KL treatment. Source data are provided as a Source Data file.

treatment were relatively evenly distributed across the three clusters of genes that we identified in keratinocyte differentiation (Fig. 2j), in contrast to the strong enrichment in "Cluster I" with the genes altered in 3-hour KL treatment. On average, the genes altered with 24-hour KL treatment featured less Pol II enrichment compared to the upregulated genes in 3-hour KL treatment. The upregulated genes with 24-hour KL treatment on average were associated with minimal Pol II pause release in differentiation, while downregulated genes were associated with an overall decrease in Pol II enrichment in differentiation (Fig. 2k, l). Similar to the previous findings in 293 T cells[18], KL treatment for 24 hours in keratinocytes reduced the protein levels of AFF1 and AFF4 (Supplementary Fig. 2e–h). Taken together, these data suggest that the intact function of the SEC is essential for progenitor maintenance, and SEC disruption progressively upregulates terminal differentiation and diminishes proliferation.

## AFF1, but not AFF4, represses epidermal differentiation

To further investigate how the SEC enforces epidermal progenitor maintenance, we explored its mutually exclusive scaffolding proteins, AFF1 and AFF4, using shRNA-mediated knockdown. Two shRNAs independently targeting AFF1 or AFF4 led to significant downregulation at both the mRNA and the protein levels in keratinocytes (Fig. 3a–d and Supplementary Fig. 3a, b). Similar to the observations from 24-hour KL treatment, both AFF1 and AFF4 knockdown decreased DNMT1 and MYC at the protein level (Supplementary Fig. 3c–f). To functionally assess how AFF1 and AFF4 influenced epidermal progenitor self-renewal capacity, we performed clonogenicity assays in a keratinocyte-3T3-cell co-culture system[22], as well as epidermal tissue regeneration experiments[23]. AFF1 or AFF4 knockdown both resulted in a significant reduction of keratinocyte clonogenicity (Fig. 3e–h). We next leveraged a two-color progenitor competition assay[24] which involves mixing 50% GFP-labeled and 50% DsRed-labeled keratinocytes to initiate epidermal regeneration in organotypic culture. This assay enables the comparison of regenerative capacity between two groups of keratinocytes within the same three-dimensional tissue. While keratinocytes expressing a non-targeting control shRNA showed similar regenerative capacity when labeled with GFP or DsRed, we found that keratinocytes with AFF1 or AFF4 knockdown had strongly decreased regenerative capacity as compared to the control (****$p < 0.001$, unpaired $t$ test, Fig. 3i, j, Supplementary Fig. 3g, h). In addition, we noticed that cells with AFF1 knockdown often accumulated on the top layer of the regenerated epidermal tissue (Fig. 3i, Supplementary Fig. 3g). In parallel to the two-color competition assay, we also generated epidermis organotypically using keratinocytes with AFF1 knockdown or AFF4 knockdown, in comparison with the non-targeting control. While both AFF1 and AFF4 impaired the formation of a full-thickness epidermis, AFF1 appeared to have a stronger effect (Supplementary Fig. 3i, j). These data suggest that AFF1 and AFF4 could play non-identical roles in regulating keratinocyte gene expression, although both are essential for supporting the progenitor function in tissue regeneration.

To explore the roles of AFF1 and AFF4 in regulating keratinocyte gene expression in more detail, we performed RNA-seq. We identified a total of 2564 differentially expressed genes with AFF1 knockdown and

1380 genes with AFF4 knockdown (fold change ≥2, $p < 0.05$, two-tailed, Wald test, Supplementary Data 4). These two data sets shared only 612 genes, with 68% of them being downregulated (Fig. 3k). The top GO terms of these shared genes included "DNA replication" and "cell division" (Fig. 3l), indicating that AFF1 and AFF4 are both required for sustaining proliferation. Genes regulated exclusively by AFF1, but not AFF4, were strongly associated with the GO terms of "epidermal development" and "keratinocyte differentiation" (Fig. 3m and Supplementary Fig. 3k). Using qRT-PCR, we validated that AFF1 knockdown, but not AFF4 knockdown, led to significant upregulation of key differentiation-activating TFs including GRHL3, OVOL1, PRDM1, and ZNF750 (Fig. 3n, o). These data demonstrate that AFF1 plays a distinct role from AFF4 in suppressing epidermal differentiation.

We further compared the RNA-seq data of AFF1 or AFF4 knockdown with the 24-hour KL treatment. Principal component analysis showed that the KL RNA-seq data clustered closely with the AFF1 knockdown data, but separately from the AFF4 knockdown data (Fig. 3p), indicating that KL treatment and AFF1 knockdown similarly affected their target genes. We found that 32% of AFF1 upregulated genes were also upregulated with 24-hour KL treatment when a two-fold cutoff was applied to both data sets (Supplementary Fig. 3l). Many of the 24-hour KL non-overlapping genes were also increased with AFF1, just not meeting the twofold cutoff (Supplementary Fig. 3m). We reasoned that if KL was indeed acting specifically on AFF1 to induce expression of differentiation activators, then in the context of AFF1 knockdown where there was little AFF1 to act on, KL treatment would show minimal additional upregulation of these genes. In the non-targeting control condition, KL induced expression of differentiation activators (Fig. 3q); in AFF1 knockdown cells, KL induced minimal changes to gene expression (Fig. 3r, s, Supplementary Fig. 3n, o). Altogether these results indicate that SEC-PTEFb suppresses epidermal differentiation in progenitors specifically through the AFF1 scaffold.

## AFF1 associates with inactive CDK9 to repress expression

With both KL treatments and AFF1 knockdown demonstrating a repressive role for AFF1-SEC in progenitor maintenance, we searched for a mechanistic explanation. We first explored the role of the CDK9 kinase activity, as part of SEC, in KL-mediated induction of differentiation-activating TFs. CDK9 inhibition alone, using two different inhibitors, flavopiridol or NVP2[25], did not mimic KL treatment in upregulating differentiation activators (Fig. 4a), suggesting that the upregulation of differentiation by KL was not a result of inhibiting CDK9 activity. Remarkably, CDK9 inhibition together with KL treatment fully blocked KL-mediated upregulation of differentiation (Fig. 4b). Similarly, CDK9 inhibition in combination with AFF1 knockdown blocked the upregulation of differentiation activators (Supplementary Fig. 4a, b). These results suggest that KL treatment upregulates differentiation gene expression through the activity switch of CDK9 from an inactive to an active state.

CDK9 activity is tightly regulated in vivo, and the inactive state of CDK9 requires the binding of HEXIM1. To further determine how CDK9 activity was involved in the repression of keratinocyte differentiation, we knocked down HEXIM1 in keratinocytes cultured in undifferentiation conditions. We designed and validated two shRNAs, and both

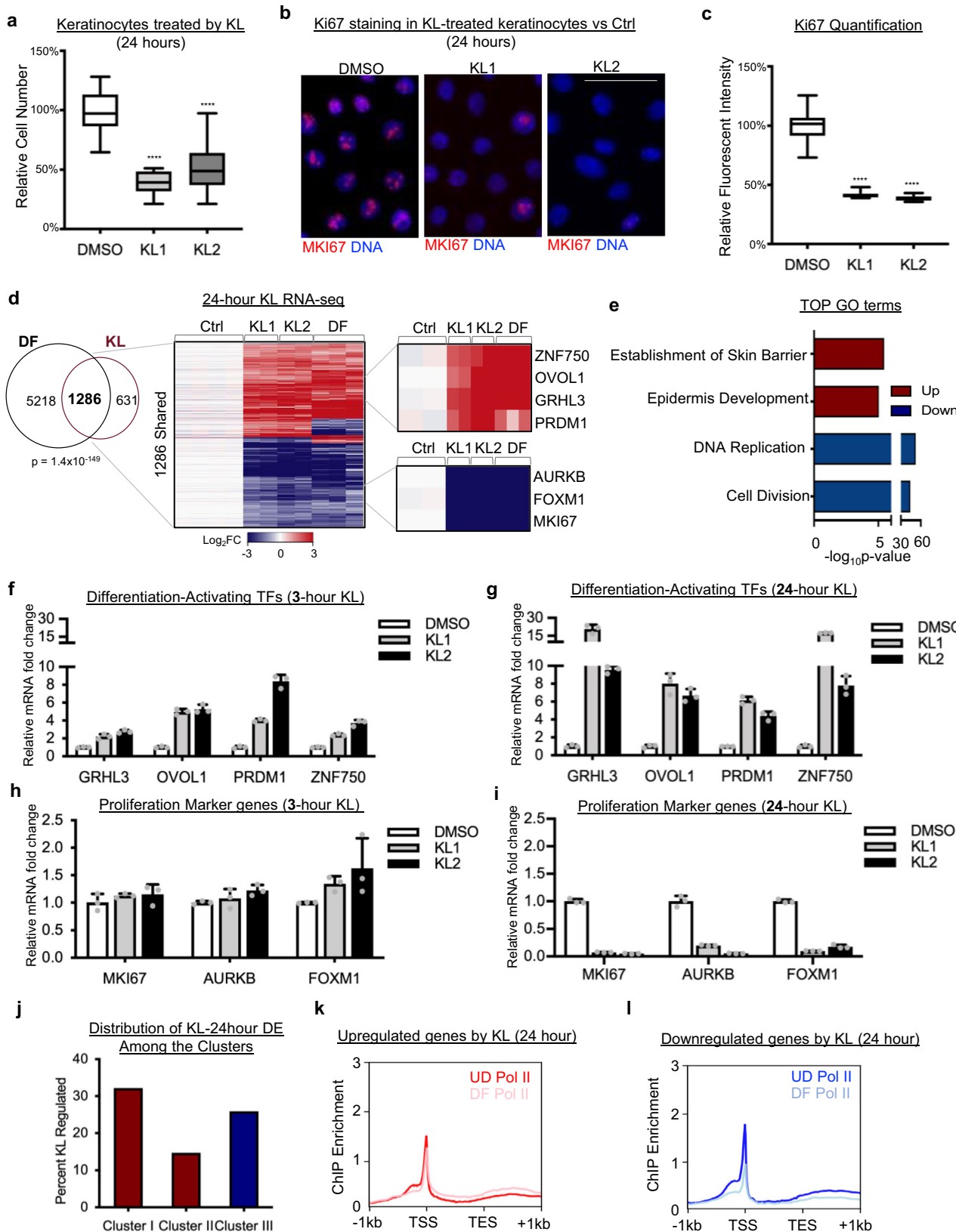

effectively knocked down HEXIM1 at the mRNA and protein levels (Fig. 4c, Supplementary Fig. 4c, d). Similar to AFF1 knockdown, HEXIM1 knockdown reduced keratinocyte clonogenicity and induced expression of differentiation-activating TFs (Fig. 4d–f). Upregulation of differentiation induced by HEXIM1 knockdown was also CDK9 dependent (Supplementary Fig. 4e, f). To assess the role of HEXIM1 in working

with the SEC, we asked if KL could still upregulate differentiation to a similar extent with HEXIM1 knockdown as compared to the non-targeting control condition. Non-targeting control cells treated with KL showed expected upregulation of differentiation relative to the DMSO control (Fig. 4g). With less HEXIM1 to act on in HEXIM knockdown conditions, the upregulation of differentiation activators was

**Fig. 2 | KL treatment progressively upregulates differentiation-activating TFs and impairs proliferation. a** Comparison of keratinocyte proliferation between DMSO control and KL treatment ($n = 15$ images, ****$P < 0.0001$, two-tailed, unpaired $t$ test, box plot represents first through third quartiles, minima, and maxima). **b** Representative Ki67 Immunofluorescent (IF) images of keratinocytes treated with DMSO control, KL1, or KL2 for 24 hours (scale bar = 50 μm). **c** Quantification of Ki67 level between DMSO control versus KL-treated keratinocytes ($n = 16$ images, ****$P < 0.0001$, two-tailed, unpaired $t$ test, box plot represents first through third quartiles, minima, and maxima). **d** Venn diagram and heatmap showing the overlap of significantly changed genes (fold change ≥2, $P < 0.05$, two-tailed, Wald test) in keratinocytes treated with KL and the calcium-induced keratinocyte differentiation signature (DF) (Fisher's exact test, Two-Tail, $P = 1.4 \times 10^{-149}$). Examples of

upregulated differentiation-activating transcription factors (TFs) and down-regulation of proliferation markers, shared in both conditions, are included in the right panel. **e** Bar graph showing the top Gene Ontology (GO) terms associated with upregulated (red) and downregulated (blue), in the overlapping, significantly changed genes in differentiation and in KL treatment (two-tailed, Fisher's exact test). **f–i** qRT-PCR comparing mRNA levels of proliferation marker genes or differentiation-activating TFs between DMSO control and KL1 or KL2 after 24 or 3 hours of KL treatment ($n = 3$ technical replicates, data are presented as mean values ± standard deviation). **j** Percent of genes in clusters I, II, and III that are differentially expressed with 24-hour KL treatment. **k, l** Average profile plots comparing Pol II ChIP-seq enrichment in the upregulated or downregulated genes after 24 hours KL treatment. Source data are provided as a Source Data file.

reduced with KL in comparison with DMSO (Fig. 4h, i, Supplementary Fig. 4g, h). These data suggest that KL-mediated upregulation of differentiation-activating TFs involves the action of HEXIM1.

To determine the full spectrum of gene expression influenced by HEXIM1 knockdown, we performed RNA-seq. HEXIM1 knockdown significantly altered 2932 genes in keratinocytes cultured in undifferentiation conditions (fold change ≥2, $p < 0.05$, two-tailed, Wald test, Supplementary Data 5). 1519 (52%) of these genes overlapped with the genes differentially expressed with AFF1 knockdown (Fig. 4j). 99% of these overlapping genes changed in the same direction with AFF1 or HEXIM1 knockdown (Fig. 4k). The shared upregulated genes were highly enriched with GO terms such as "epidermis development" and "keratinization", while the downregulated genes were related to "cell division" and "DNA replication" (Fig. 4l). Given the similar roles of AFF1 and HEXIM1 in regulating keratinocyte gene expression in the progenitor state, we asked whether they directly interact with each other. We expressed HA-tagged AFF1 in keratinocytes and performed co-immunoprecipitation experiments. In keratinocytes cultured in the progenitor state, but not in the differentiated state, HA-AFF1 efficiently co-immunoprecipitated HEXIM1. AFF1's interactions with CDK9 as well as P-CDK9 (phosphorylated threonine 186) were detected in both the undifferentiated and the differentiated states (Fig. 4m, n). Thus, these data indicate that HEXIM1 interacts with AFF1-SEC to synergistically suppress differentiation genes by maintaining CDK9 in an inactive state, specifically in the progenitor state but not in the differentiated state of human keratinocytes.

### HEXIM1 and AFF1 directly suppress rapid-response targets

To further explore how AFF1 and HEXIM1 cooperate to suppress gene expression, we performed AFF1 and HEXIM1 ChIP-seq in undifferentiated keratinocytes. Among the total 3338 HEXIM1 ChIP-seq peaks, we found that 2112 (63%) overlapped with AFF1 peaks (Fig. 5a). These shared peaks overall had higher Pol II enrichment than the non-overlapping regions (****$p < 0.001$, unpaired $t$ test, Fig. 5b). The majority of these HEXIM1-AFF1 shared peaks were enriched in the promoter-proximal regions of genes (TSS ± 1 kb) where there was strong co-localization of AFF1 and HEXIM1 ChIP-seq signal (Fig. 5c, d). We also performed CDK9 ChIP-seq using HA-tagged CDK9. We found that 93% of AFF1-HEXIM1 shared peaks overlap with CDK9 ChIP-seq peaks (Supplementary Fig. 5a–c). These data indicate that AFF1, HEXIM1, and CDK9 co-occupy the promoter-proximal regions of their shared target genes, which also feature high enrichment of Pol II binding.

We then leveraged this ChIP-seq information to more closely investigate the genes induced by the 3-hour KL treatment. This led us to identify 92 "rapid-response" genes, which were rapidly induced by KL and were direct targets of both AFF1 and HEXIM1 (Fig. 5e, Supplementary Data 6). This set of 92 genes accounted for 52% of total 3-hour KL upregulated genes, while only 14% of 24-hour KL differentially expressed genes were direct targets of HEXIM1 and AFF1 (Supplementary Fig. 5d). Among these "rapid-response" genes, the two most highly upregulated genes by 3-hour KL were ATF3 and DUSP1 (Fig. 5f).

This list also included RND3/RhoE, which had been implicated in influencing keratinocyte differentiation[26]. All three of these genes were upregulated at the protein level with 3-hour KL treatment (Supplementary Fig. 5e–h). To better characterize the kinetics of transcriptional activation at rapid-response genes compared to differentiation activators, we performed nascent RNA-seq at 1-hour and 3-hour time points with KL treatment. Consistent with the trend we observed with RNA-seq using purified mRNA, more genes were upregulated than downregulated with KL at both the earlier time point (1 hour) and the later time point (3 hour) with nascent RNA sequencing (Supplementary Fig. 5i.). With these data, we found that ATF3, DUSP1, and RND3 were all upregulated prior to the differentiation activators (Fig. 5g). These three "rapid-response" genes were also upregulated in the context of AFF1 and HEXIM1 knockdown as well as in keratinocyte differentiation (Fig. 5h, i and Supplementary Fig. 5j). In addition to having AFF1, HEXIM1, and CDK9 ChIP-seq peaks near the transcription start sites, ATF3, DUSP1, and RND3 also had high promoter-proximal Pol II occupancy in the progenitor state, and a Pol II pause release signature in the differentiated keratinocytes (Fig. 5j–l). There was, however, minimal AFF1 and HEXIM1 enrichment at the differentiation activators, suggesting that the rapid responding direct targets may play upstream roles to the differentiation activators (Supplementary Fig. 5k). Thus, our integrated analysis of ChIP-seq and RNA-seq identified these three "rapid-response" genes as candidates that might further advance keratinocyte differentiation.

Key genes related to progenitor self-renewal, MYC, and DNMT1, also feature AFF1 ChIP-seq peaks near their transcription start sites, suggesting that the intact function of AFF1 is also involved in directly sustaining the expression of these genes (Supplementary Fig. 5l). However the expression of these genes was not significantly altered within 3 hours of KL treatment, indicating that the downregulation of these genes was not part of the rapid-response (Supplementary Fig. 5m).

### Rapid-response gene ATF3 drives keratinocyte differentiation

To determine if the upregulation of ATF3, DUSP1, or RND3, induced by 3-hour KL treatment, was sufficient to promote keratinocyte differentiation, we overexpressed these genes using a doxycycline-inducible system in keratinocytes cultured in undifferentiation condition. The overexpression of these three candidate genes was validated by western blotting (Fig. 6a–c). Of the three candidates, only ATF3 overexpression was sufficient to significantly induce the expression of differentiation activators GRHL3, OVOL1, ZNF750, and PRDM1 (Fig. 6d–f). Consistently, the overexpression of ATF3, but not DUSP1 or RND3, resulted in the loss of keratinocyte clonogenicity (Fig. 6g–j). We further evaluated the functional impact of ATF3 overexpression in epidermal tissue regeneration. ATF3 overexpression resulted in hypoplasia, with significantly decreased epidermal thickness as compared to control (Supplementary Fig. 6a, b). These findings identified ATF3, suppressed directly by AFF1 and HEXIM1, as an early and potent initiator to further promote the expression of other differentiation-activating TFs.

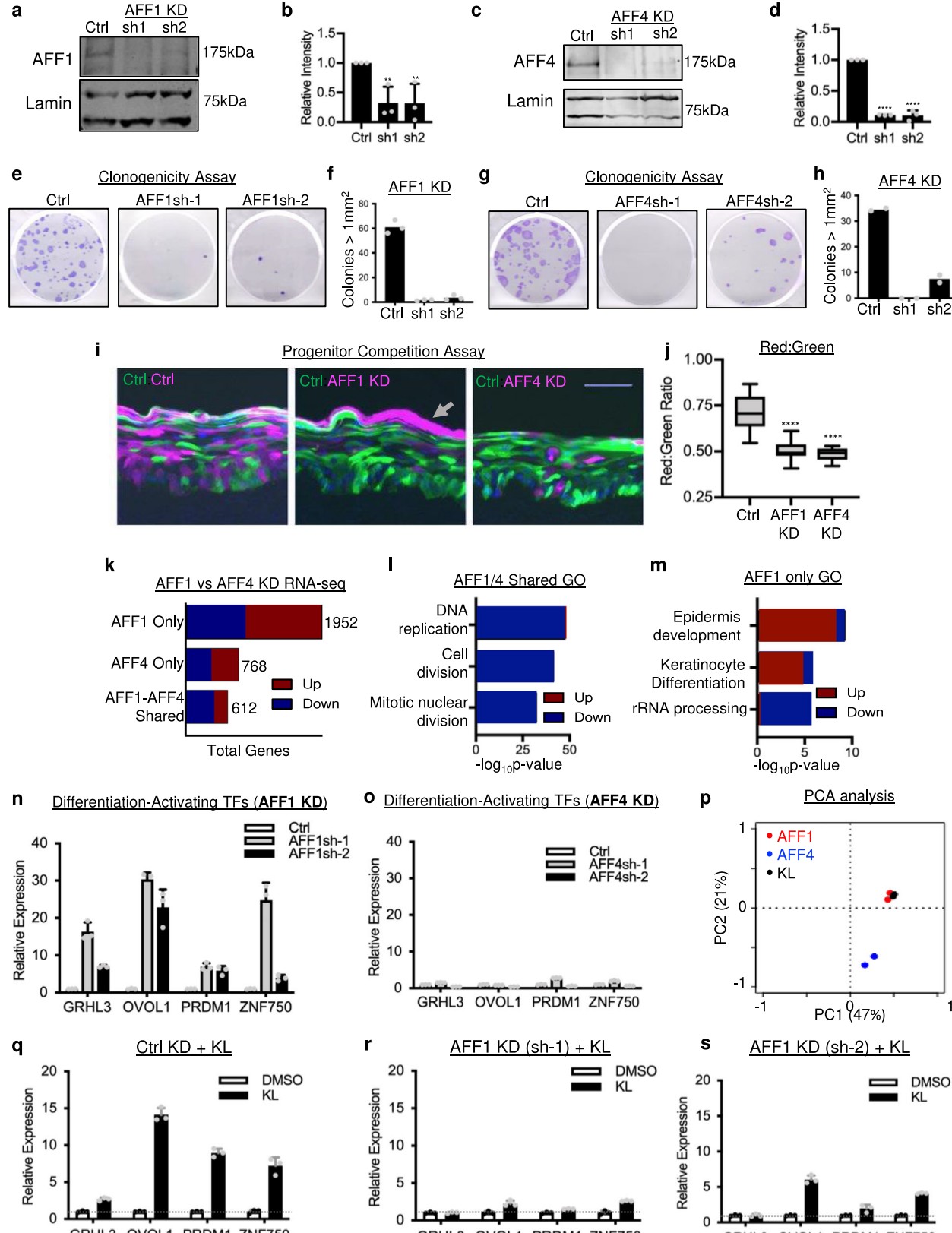

## PKC signaling triggers rapid induction of SEC targets

Given the critical roles of SEC in suppressing the differentiation-initiators such as ATF3 in the progenitor state keratinocytes, as well as the rapid induction of SEC targets through CDK9 activation, we asked if this mechanism of CDK9 activity switch is utilized by specific signaling pathways to initiate keratinocyte differentiation. Elevated calcium concentration (1.2 mM) in combination with confluency is a commonly-used approach to induce keratinocyte differentiation[1,24,27,28]. The "rapid-response" genes were upregulated with this approach after 96 hours of treatment to a similar extent as 3 hours of KL treatment (Supplementary Fig. 7a). We further investigated if elevated calcium concentration alone in keratinocytes cultured in sub-confluent conditions could

**Fig. 3 | SEC-scaffold, AFF1, but not AFF4, is necessary for repressing epidermal differentiation. a–d** Western blots showing knockdown (KD) efficiency of the shRNA's targeting AFF1 or AFF4. ($n$ = 3 biological replicates, AFF1 sh1 $P$ = 0.0131, AFF1 sh2 $P$ = 0.0227, AFF4 sh1 $P$ < 0.0001, AFF4 sh2 $P$ < 0.0001, two-tailed, unpaired $t$ test, data are presented as mean values ± standard deviation). **e–h** Representative images and quantification of clonogenic assays comparing keratinocytes expressing shRNA's targeting AFF1 or AFF4, versus non-targeting control shRNA ($n$ = 2–3 technical replicates). **i, j** Representative images and red:green fluorescence-quantification of epidermal tissue sections. Each piece of epidermal tissue was regenerated with 50% of GFP-labeled keratinocytes expressing non-targeting control shRNA and 50% of DsRed-labeled keratinocytes expressing non-targeting control shRNA (left), AFF1-targeting shRNA 1 (middle) or AFF4-targeting shRNA 1 (right). DsRed-labeled cells are represented in magenta. (scale bar = 100 μm, $n$ = 15, ****$P$ < 0.0001, two-tailed, unpaired $t$ test, box plot represents first through third quartiles, minima, and maxima). **k** Comparison of genes differentially expressed in RNA-seq with AFF1 KD, AFF4 KD, and in both AFF1 and AFF4 KD conditions. **l** Top Gene Ontology (GO) terms of genes differentially expressed in both AFF1 and AFF4 KD RNA-seq data sets (two-tailed, Fisher's exact test). **m** Top GO terms of genes differentially expressed in AFF1 but not AFF4 KD RNA-seq (two-tailed, Fisher's exact test). **n, o** qRT-PCR comparing the expression of differentiation-activating TFs between non-targeting control and AFF1 or AFF4 KD ($n$ = 3 technical replicates, data are presented as mean values ± standard deviation). **p** Principal Component Analysis (PCA) of AFF1 KD, AFF4 KD, and KL 24-hour RNA-seq data sets. **q–s** qRT-PCR comparing mRNA levels of differentiation-activating TFs between DMSO control and KL2 treatment with non-targeting control or AFF1 knockdown using two independent shRNA's. Control non-targeting KL is relative to non-targeting DMSO, AFF1 sh1 KL is relative to AFF1 sh1 DMSO, and AFF1 sh2 KL is relative to AFF1 sh2 DMSO ($n$ = 3 technical replicates, data are presented as mean values ± standard deviation). Source data are provided as a Source Data file.

be sufficient to rapidly induce the expression of these genes in sub-confluent keratinocytes, but only minimal changes were observed (Supplementary Fig. 7b). These findings suggest that an alternative signaling process, rather than elevated calcium concentration alone, could be utilizing the CDK9 activity switch to promote differentiation.

PKC signaling has been previously implicated in promoting keratinocyte differentiation[29]. Furthermore, PKC's substrates include HEXIM1, and the phosphorylation of HEXIM1 mediated by PKC can alter HEXIM1's association with CDK9[30,31]. We, therefore, tested if TPA, a PKC activator, could induce the rapid upregulation of SEC direct targets. As early as 1 hour after treatment, TPA strongly induced the upregulation of ATF3, DUSP1, and RND3. Notably, this rapid induction appeared to be transient at the mRNA level, as the fold change of these targets decreased after 3 hours of TPA treatment (Fig. 7a). At the protein level, the upregulation of these key differentiation activators such as ATF3 was sustained with 3-hour TPA treatment (Supplementary Fig. 7c, d). Differentiation-activating TFs were also induced at the 1-hour time point, and the expression of these TFs continued to rise at the 3-hour time point by TPA (Fig. 7a). TPA treatment resulted in no rapid change to cell-cycle gene expression (Supplementary Fig. 7e), similar to the observations from the short-term (3 hour) KL treatment. Since PKC signaling can further activate other kinases downstream, we focused on the changes detectable within 1 hour of TPA treatment to minimize the influences from potential indirect effects. We further validated that the gene upregulation observed with TPA treatment was indeed a result of PKC activation, by pretreating keratinocytes with two different PKC inhibitors, Gouml6983 (Gou) or bisindolylmaleimide I (Bis). When keratinocytes were pretreated with these PKC inhibitors, TPA was no longer able to induce these "rapid-response" genes or differentiation-activating TFs (Fig. 7b). These findings unveiled a strong similarity, between PKC activation and KL treatment in controlling a subset of SEC direct targets as well as differentiation-activating TFs.

We then asked if the rapid induction of these genes with PKC activation was conferred through the SEC. If TPA activated these genes through the SEC, we posited that after KL treatment TPA would not cause any further upregulation. Indeed, after keratinocytes were treated with KL, we observed no further drastic changes in these genes with TPA treatment (Fig. 7c, d, Supplementary Fig. 7f). Similarly, when we knocked down AFF1 or HEXIM1 and treated with TPA, TPA induced only minimal upregulation beyond what was already seen in knockdown (Supplementary Fig. 7g–j). These data demonstrate that TPA induces differentiation by acting on the repressive AFF1-HEXIM complex. When that complex is impaired through KL treatment or AFF1/HEXIM knockdown, TPA could no longer drastically upregulate the expression of SEC targets and the differentiation activators, as CDK9 would already have been released to an active state. Consistent with this model, we found that this TPA-mediated rapid upregulation of differentiation-activating TFs was also dependent on CDK9 activity. Similar to KL treatment, CDK9 kinase inhibition completely abolished the upregulation of these genes by TPA (Fig. 7e). To more broadly compare the gene expression changes observed with KL treatment to those of TPA treatment, we performed RNA-seq after 1-hour TPA treatment and compared the data to 3-hour KL RNA-seq (fold change ≥ 2, $p$ < 0.05, two-tailed, Wald test Supplementary Data 7). 49.7% of the differentially expressed genes with 1-hour TPA treatment overlapped with those from 3-hour KL treatment. These overlapped genes were almost exclusively upregulated in both conditions (Fig. 7f), suggesting that the early response of PKC signaling involves the activation of a subset of SEC targets. Taken together, our findings support a model that the activity switch of SEC-CDK9, utilized by signaling pathways such as PKC, plays a crucial role in rapidly activating a subset of differentiation initiators, which further promote the expression of differentiation-activating TFs to advance the terminal differentiation process.

## CDK9 is dissociated from HEXIM1 upon KL/TPA treatment

Given the activity switch of CDK9 observed with KL or TPA treatment, we asked if these treatments disrupted CDK9 and HEXIM1 association. In undifferentiated keratinocytes treated with DMSO as control, HA-tagged CDK9 co-immunoprecipitated with HEXIM1; however, this association with HEXIM1 was lost in the context of either 3-hour KL or 1-hour TPA treatment (Fig. 8a–c). Building on this, we performed ChIP-seq with HA-tagged CDK9 to assess potential changes of CDK9 chromatin binding with the treatment of KL or TPA. We calculated a traveling ratio to determine the enrichment of CDK9 in the gene body relative to the promoter of the 92, direct, rapid-response genes we identified. The CDK9 traveling ratio significantly increased with both KL and TPA treatment (Fig. 8d–g), indicating an increased proportion associated with the gene bodies. This trend is clearly visible in ATF3 as an example, from the genome browser tracks of these CDK9 ChIP-seq data sets (Fig. 8h). Thus, these data indicate that KL or TPA treatment breaks CDK9 from its association with HEXIM1, releasing CDK9 from promoter-proximal binding and allowing CDK9 to engage in activating gene transcription.

Given the drastic differences in Pol II binding with these rapid-response genes between the undifferentiated and the differentiated states, we asked if this trend could be mimicked with KL or TPA treatment. Interestingly, only subtle changes were observed comparing Pol II ChIP-seq in keratinocytes treated with KL for 1 hour or 3 hours relative to the DMSO control (Supplementary Fig. 8a–c). Similar results were obtained comparing 0.5 or 1 hr TPA treatment with DMSO (Supplementary Fig. 8d–f). These subtle changes of Pol II binding were visible in the genome browser tracks of ATF3, showing an increase in Pol II along the gene body, especially under 1-hour KL and 30min-TPA conditions (Supplementary Fig. 8g). These data suggest that the treatment using KL or TPA "unstablizes" Pol II from the paused state;

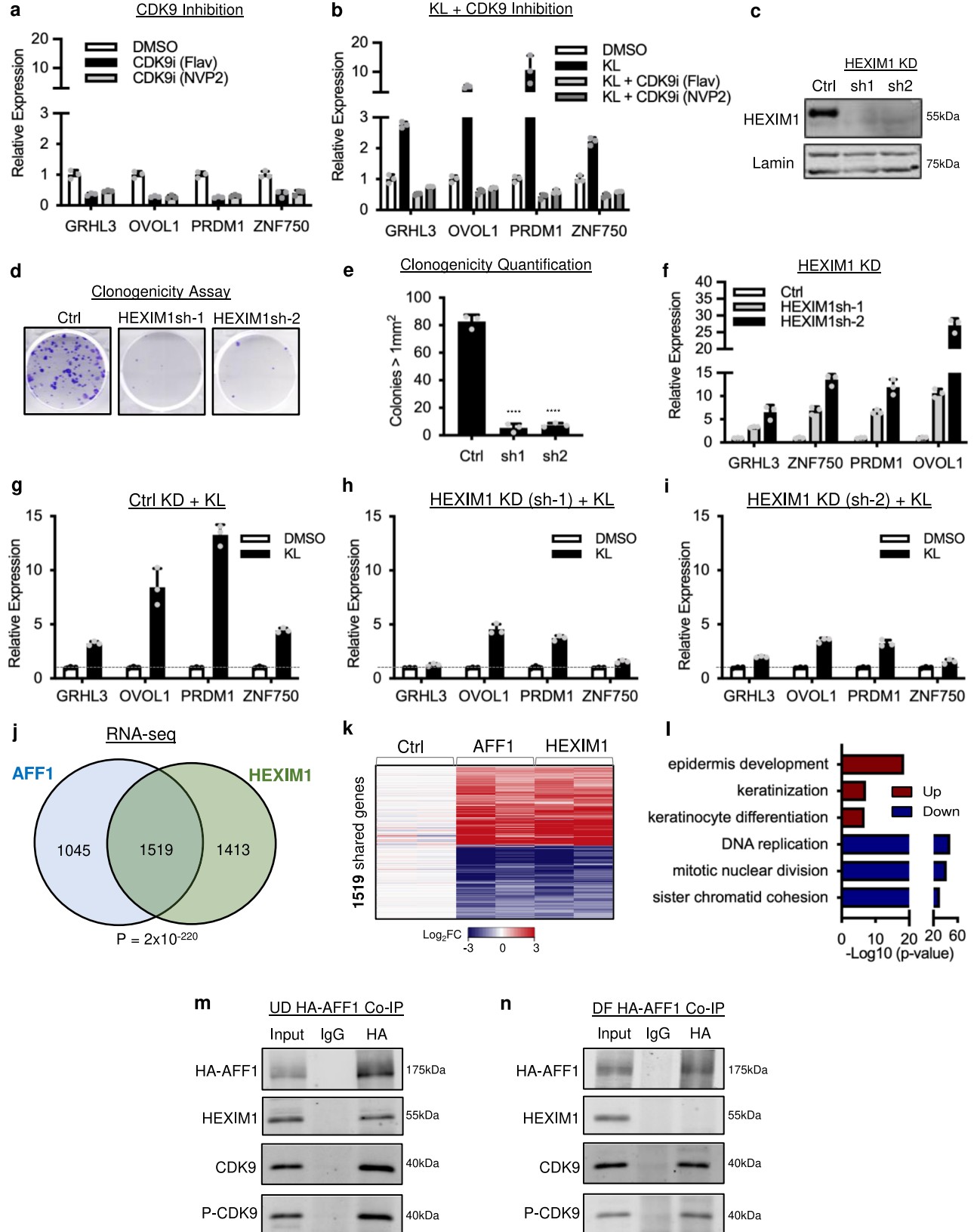

the stabilized elongation state as observed in the differentiated keratinocytes (with 4 days of Ca²⁺ treatment in combination with confluency) may require other mechanisms working in conjunction with the initial action from SEC.

We also asked how chromatin binding of the AFF1-HEXIM repressive complex changed upon SEC perturbation and with PKC activation. Using ChIP-qPCR, we observed that AFF1 enrichment decreased with both KL and TPA treatment while HEXIM1 binding remained generally the same (Supplementary Fig. 8h, i).

In addition, we asked what could be activating CDK9 once it is released from the inactive complex. Given that BRD4 is one of the well-established activators of CDK9 and BRD4 activity is required for

**Fig. 4 | AFF1 associates with inactive CDK9 to repress target gene expression.**
**a** qRT-PCR showing mRNA levels of differentiation activators with 3-hour CDK9 inhibitor (CDK9i) treatment with flavopiridol (Flav) or NVP2 relative to DMSO control ($n = 3$ technical replicates, data are presented as mean values ± standard deviation). **b** qRT-PCR showing the expression of differentiation-activating TFs in keratinocytes treated with KL alone or in combination with CDK9 inhibitors, as compared to DMSO control ($n = 3$ technical replicates, data are presented as mean values ± standard deviation). **c** Western blot showing knockdown (KD) efficiency of HEXIM1 shRNA's ($n = 3$ biological replicates, quantified in Supplementary Fig. 4c). **d** Clonogenic assay of human keratinocytes expressing HEXIM1-shRNA's or non-targeting control shRNA. **e** Quantification of colonies >1 mm$^2$ in HEXIM1 KD conditions relative to control ($n = 3$/group, ****$P < 0.0001$, two-tailed, unpaired $t$ test, data are presented as mean values ± standard deviation). **f** qRT-PCR comparing the mRNA levels of differentiation activators in HEXIM1 KD relative to control ($n = 3$ technical replicates, data are presented as mean values ± standard deviation).

**g**–**i** qRT-PCR showing the relative expression of differentiation-activating TFs between DMSO control and KL treatment where control non-targeting KL is relative to non-targeting DMSO, HEXIM1 sh1 KL is relative to HEXIM1 sh1 DMSO, and HEXIM1 sh2 KL is relative to HEXIM1 sh2 DMSO ($n = 3$ technical replicates, data are presented as mean values ± standard deviation). **j** Venn diagram comparing genes differentially expressed in AFF1 KD versus HEXIM1 KD RNA-seq (Fisher's exact test, two-Tail, $P = 2 \times 10^{-220}$). **k** Heatmap showing shared genes significantly changed (fold change ≥2, $P < 0.05$, two-tailed, Wald test) in AFF1 KD RNA-seq or HEXIM1 KD RNA-seq. **l** Top Gene Ontology terms of the genes differentially expressed in AFF1 and HEXIM1 KD RNA-seq (two-tailed, Fisher's exact test). **m**, **n** Co-immunoprecipitation of HA-AFF1 western blot showing interactions between AFF1 and HEXIM1, CDK9, and phosphorylated threonine 186 CDK9 (P-CDK) in progenitors (UD) and in differentiated (DF) keratinocytes ($n = 3$ biological replicates). Source data are provided as a Source Data file.

inducing epidermal terminal differentiation in calcium-mediated differentiation condition[32], we tested if BRD4 is involved in the transition of CDK9 from an inactive to an active state with KL treatment. Comparing keratinocytes treated with KL to those treated with KL in combination with a BRD4 inhibitor (JQ1 or CPI203), we found that the upregulation of differentiation activators was dependent on BRD4 activity. However, among the three rapid-response genes tested, BRD4 inhibition only impaired the induction of RND3 but not ATF3 or DUSP1 (Supplementary Fig. 8j). Thus, more exploration will be required to answer if other proteins are necessary for CDK9 activity at rapid-response genes directly controlled by AFF1 and HEXIM1, although CDK9 can be facilitated by BRD4 to induce expression of differentiation activators that are downstream to the rapid-response genes.

Taken together, our findings highlight a crucial role of AFF1 and HEXIM1 in influencing epidermal progenitor fate decision. We found that both AFF1 and HEXIM1 directly suppress a group of rapid-response genes, which features a high level of paused Pol II in the progenitor state. Both AFF1 and HEXIM1 are essential for maintaining CDK9 in its inactive form in this state. With the disruption of AFF1/HEXIM1 function or upon PKC activation, CDK9 kinase activity is switched on with dissociation from HEXIM1. This initiates the differentiation cascades by firstly upregulating the direct rapid-response genes such as ATF3, which further promotes the upregulation of differentiation-activating transcription factors including OVOL1, GRHL3, ZNF750, and PRDM1 (Fig. 8i).

## Discussion

In this study, we identified a crucial role of CDK9 activity switch, modulated by AFF1 and HEXIM1, in progenitor fate decisions between self-renewal and differentiation. CDK9 activity influences Pol II dynamics, and we began by performing a clustering analysis of Pol II binding in association with gene expression in keratinocyte differentiation. In agreement with the findings from several other stem-cell differentiation processes, including that of the erythropoietic progenitors[33], neuronal progenitors[34], and mouse embryonic stem cells[35], we found that Pol II pause release is associated with a subset of upregulated genes in keratinocyte differentiation. In particular, we identified a unique cluster of upregulated genes ("cluster I") that feature the highest enrichment of Pol II, and the Pol II binding transitions from paused to elongation in differentiation. As Pol II is already pre-recruited to the promoter-proximal regions, these genes are poised for rapid induction. Leveraging the newly developed SEC peptidomimetic inhibitors (KL1 and KL2), we found that a subset of "cluster I" genes are directly controlled by the SEC. With KL or TPA treatment, the upregulation of these SEC targets was among the earliest changes in the transcriptome. These findings suggest that the upregulation of these SEC targets is likely to be an early event during epidermal progenitor fate switch from self-renewal to differentiation.

Our study highlighted that the genes featuring high enrichment of paused Pol II in the progenitor state could be involved in the early initiation processes of differentiation. First, these genes only account for a small fraction of the genes that are upregulated during differentiation. Second, these genes are highly enriched in the molecular function of transcription regulation. Third, our overexpression experiment demonstrated that a subset of these genes, such as ATF3, function as differentiation initiators to promote the expression of other differentiation-activating TFs such as OVOL1. Notably, the OVOL1 genomic locus features little Pol II enrichment in the progenitor state, but robust Pol II recruitment in differentiation. These findings support a model where genes featuring high Pol II enrichment are the early responders upon receiving differentiation stimuli; these early responders function to subsequently activate the next tier of differentiation-activating TFs to advance the process of differentiation. This is reminiscent of the previous findings in *drosophila* embryonic development where the enrichment of Pol II occupancy is correlated with the timing of gene activation[36].

In the differentiated state of keratinocytes, several Pol II elongation regulators, such as BRD4, PAF1, and SPT6, have been identified as essential genes for the full induction of terminal differentiation[27,32]. In agreement with the essentiality of other elongation regulators in activating epidermal differentiation, our data indicate that the CDK9 activation in association with the SEC contributes to differentiation initiation, but is not sufficient to fully drive terminal differentiation. In keratinocytes maintained in the undifferentiated culture condition, KL or TPA treatment immobilized Pol II from the stably paused state based on our ChIP-seq data. However, this level of immobilization appears to be transient and highly dynamic, within the limited time points that we were able to experimentally capture using ChIP-seq. In the example of ATF3, we were able to detect increased Pol II occupancy in the gene body with KL or TPA treatment. However, Pol II did not reach a stable elongation state as observed in keratinocytes that were differentiated for 4 days with both high levels of calcium as well as confluency. These findings suggest that additional mechanisms are involved in fully engaging Pol II to a stable elongation state in terminal differentiation.

In the progenitor state, we identified that SEC-scaffold AFF1 interacts with HEXIM1, and both are essential for maintaining the CDK9 in the inactive state to suppress the expression of their direct target genes. Using shRNA's targeting HEXIM1, we observed strong upregulation of differentiation in keratinocytes cultured in the undifferentiation condition. These observations from HEXIM1 knockdown are in agreement with a recent paper reporting 7SK knockdown in human keratinocytes, where differentiation marker genes such as TGM and INV were upregulated[37]. Both 7SK and HEXIM1 are essential for tethering CDK9 in the inactive state, yet how CDK9 activity contributes to the differentiation process was not characterized in this previous

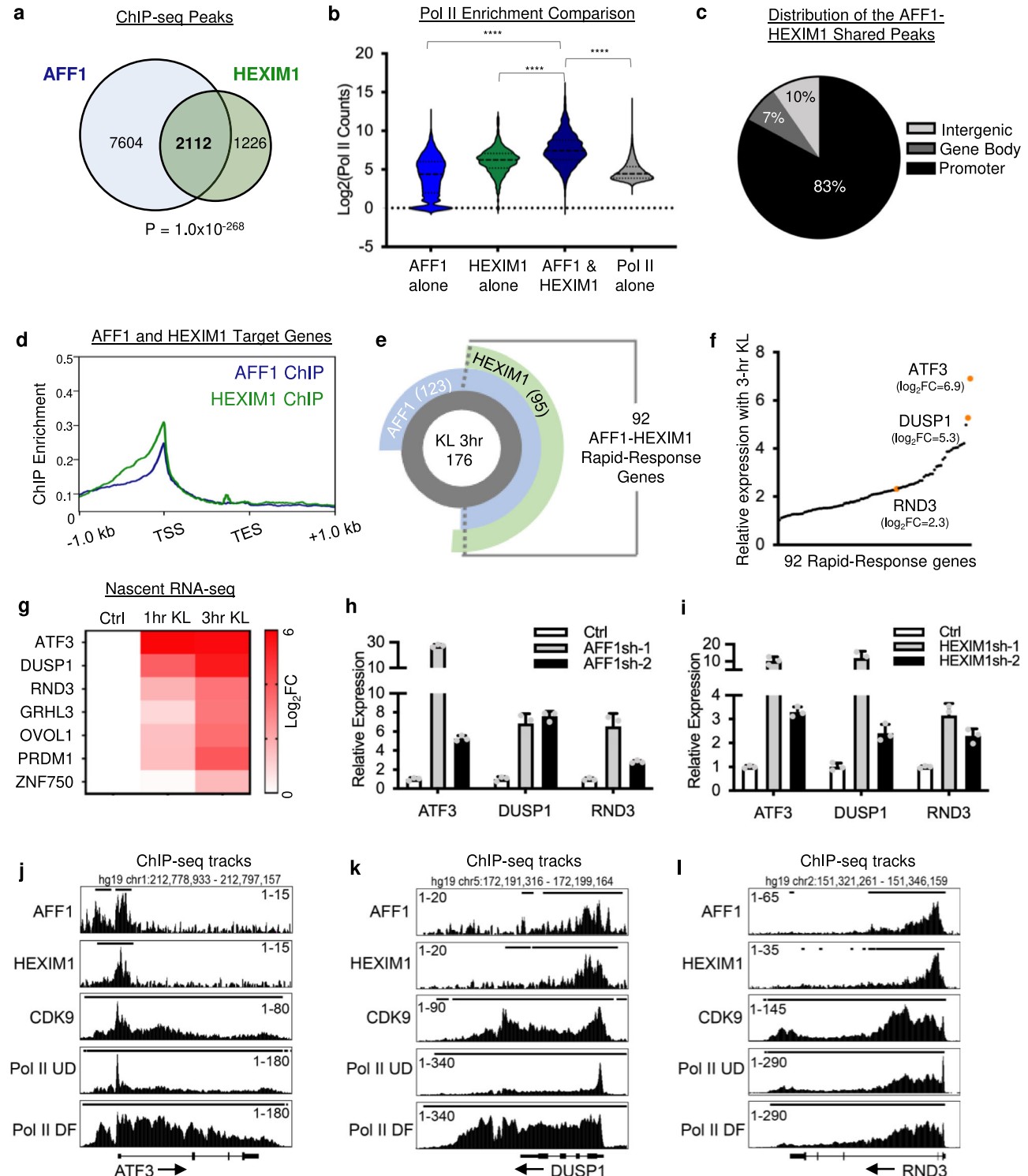

**Fig. 5 | HEXIM1 and AFF1 colocalize to directly suppress rapid-response targets.**
**a** Venn diagram showing the significant overlap between AFF1 and HEXIM1 ChIP-seq peaks (Fisher's exact test, two-tailed, $P = 1 \times 10^{-268}$). **b** Violin plot showing the significant enrichment of total Pol II counts at ChIP-seq categories: AFF1 unique peaks, HEXIM1 unique peaks, and peaks with Pol II but no HEXIM1 or AFF1 relative to overlapping AFF1 and HEXIM1 peaks (****$P < 0.0001$, two-tailed, unpaired $t$ test). **c** Pie chart showing the peak distribution of AFF1 and HEXIM1 ChIP-seq data in promoters (transcription start site (TSS) ±1 kb), gene bodies (TSS ±1 kb through transcription end site), or intergenic regions. **d** Average profile plot showing the overlapping enrichment of AFF1 and HEXIM1 ChIP-seq signal near the promoters of their shared target genes. **e** Shared target genes among AFF1 ChIP-seq, HEXIM1 ChIP-seq, and the significantly upregulated genes in keratinocytes with 3-hr KL

treatment. **f** Relative expression of the 92 AFF1-HEXIM1-KL(3 hr) shared target genes, ranked from low to high, in keratinocytes treated with KL versus DMSO for 3 hours. The top two upregulated genes ATF3 and DUSP1, as well as RND3, are highlighted in orange. **g** Heatmap showing differential expression of rapid-response and differentiation-activating genes with 1-hour and 3-hour nascent RNA-seq. **h, i** qRT-PCR showing the upregulation of ATF3, DUSP1, and RND3 with AFF1 or HEXIM1 knockdown ($n = 3$ technical replicates, data are presented as mean values ± standard deviation). **j–l** Genome browser tracks showing the enrichment of AFF1, HEXIM1, CDK9, undifferentiated (UD) Pol II, and differentiated (DF) Pol II ChIP-seq signal at ATF3, DUSP1, and RND3. Source data are provided as a Source Data file.

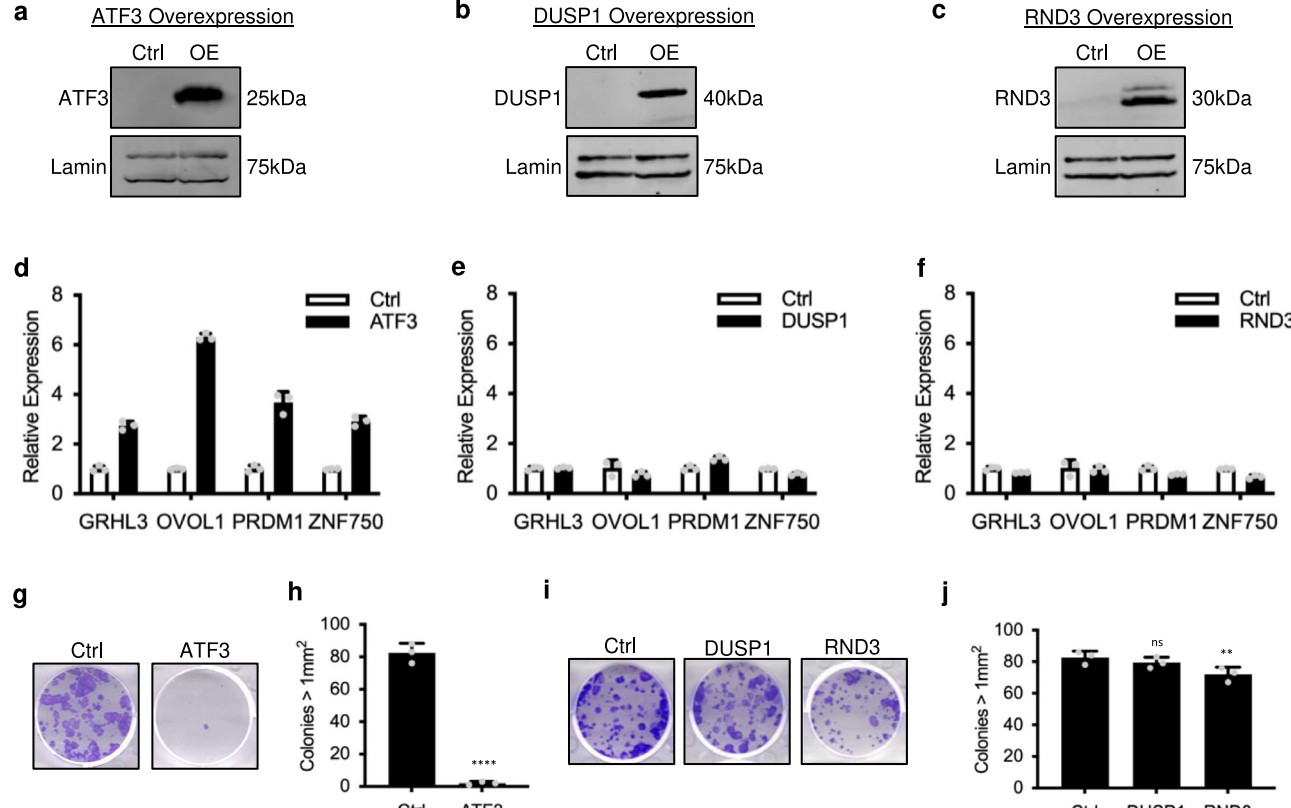

**Fig. 6 | The SEC rapid-response gene ATF3 drives keratinocyte differentiation.** **a**–**c** Western blot showing the overexpression (OE) of ATF3, DUSP1, or RND3 in keratinocytes using the pTRIPZ vector post DOX induction. **d**–**f** qRT-PCR showing the mRNA expression levels of differentiation-activating TFs with ATF3, DUSP1, or RND3 overexpression ($n = 3$ technical replicates, data are presented as mean values ± standard deviation). **g**–**j** Representative images and quantification of clonogenic assays comparing human keratinocytes expressing ATF3, DUSP1, or RND3 versus the empty-vector control. Colonies >1 mm$^2$ are included for quantification ($n = 3$/group, ****$P < 0.001$, **$P < 0.05$, ns = not significant, ATF3 $P < 0.0001$, DUSP1 $P = 0.3489$, RND3 $P = 0.0406$, two-tailed, unpaired $t$ test, data are presented as mean values ± standard deviation). Source data are provided as a Source Data file.

study. Here we demonstrate that HEXIM1 interacts and colocalizes with AFF1 in the promoter-proximal regions of their direct target genes. We further showed that CDK9 activation underlies the upregulation of AFF1-HEXIM1 direct targets with KL or TPA treatment. Interestingly, our co-immunoprecipitation results showed that AFF1-HEXIM1 association occurs in the progenitor state but not in the differentiated keratinocytes. While this study focused on the roles of AFF1 and HEXIM1 in progenitors, how AFF1 and HEXIM1 influence gene expression in differentiated keratinocytes where they no longer interact remains unclear and will be a direction for future investigation. In addition, AFF1 and HEXIM1 interaction was observed in the context of HIV transcription activation[38] indicating that the molecular interaction between AFF1-HEXIM1 is being utilized in regulating other processes in addition to modulating progenitor differentiation.

Although this study focuses on SEC's roles in repressing a subset of rapid-response genes, we don't exclude SEC's role as an activator for other targets. The gene-activating role of the SEC in stimulating Pol II elongation has been recognized in several other contexts. The reconstituted AFF4-containing SEC, or just the catalytic subunit ELL2 with its associated protein EAF1, is sufficient to stimulate robust Pol II elongation in vitro[39]. A pool of active CDK9-containing SEC is associated with the HIV transactivator protein Tat[40], and in addition, AFF4-containing SEC facilitates the activation of MLL chimeric fusions in leukemia[41]. SEC also associates with active PTEFb to directly stimulate Pol II elongation of MYC, promoting the progression of MYC-driven cancer[18]. Consistent with these previous reports, we also find MYC among the direct targets in our AFF1 ChIP-seq data in the undifferentiated keratinocytes. DNMT1, another essential regulator for

epidermal progenitor maintenance, also has AFF1 ChIP-seq enrichment near the promoter region. Both MYC and DNMT1 are downregulated at the protein level with AFF1/4 knockdown and with 24-hour KL treatment in keratinocytes; however, the mRNA expression of these genes is not significantly altered within shorter 1 and 3-hour KL treatments. In general, the short-term KL or TPA treatment resulted in predominant upregulation of genes, rather than downregulation of the direct target genes, based on our nascent RNA-seq as well as traditional RNA-seq. Thus, the downregulation of other SEC targets is just not within the first wave of gene expression changes in our system.

Among the "rapid-response" genes, ATF3 is the most highly induced target after 3 hours of KL treatment. ATF3 induction has been previously connected to several extracellular signals including UV irradiation[42] and ER stress[43]. Depending on the context, ATF3 can function as a repressor or activator of gene expression[44]. Our study has identified ATF3 induction as one of the earliest responses to PKC signaling. We further demonstrate that ATF3 functions as an activator in this context to further promote the expression of differentiation-activating TFs. In addition, the rapid-response genes identified in this study are enriched in a single-cell cluster ("BAS IV") of human epidermal cells transitioning from the progenitor state immediately to the differentiated state[45], suggesting that the induction of these rapid-response genes are involved in the process of human epidermal tissue differentiation.

PKC signaling is linked to keratinocyte differentiation in a number of the previous studies[2,29,46,47], although the exact mechanisms still remain incompletely understood owing to its complexity. A variety of PKC substrates have been identified[48], including both cytoplasmic and

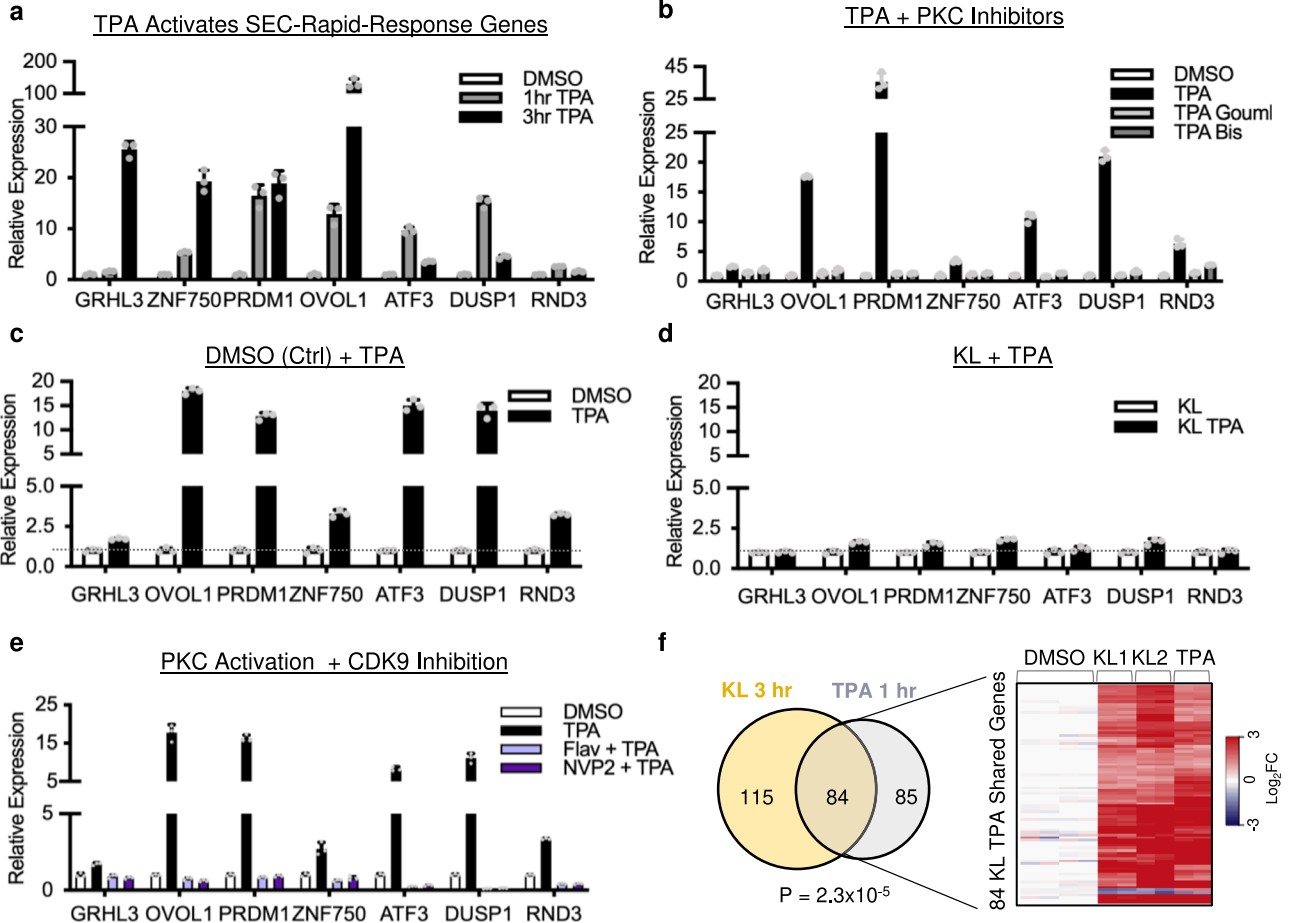

**Fig. 7 | SEC mediates the rapid initiation of differentiation in response to PKC signaling. a** qRT-PCR comparing the relative mRNA expression of differentiation-activating TFs as well as ATF3, DUSP1, and RND3 with 1 hour or 3 hours of TPA treatment relative to DMSO control ($n = 3$ technical replicates, data are presented as mean values ± standard deviation). **b** qRT-PCR shows that the PKC inhibitors, gouml6983 (Gouml) or bisindolylmaleimide I (Bis), consistently suppress the upregulation of representative genes induced by TPA treatment (1 hour) as compared with the DMSO control ($n = 3$ technical replicates, data are presented as mean values ± standard deviation). **c, d** qRT-PCR comparing gene expression induced by TPA alone or in combination with KL where panel c is TPA relative to a DMSO control and **d** is KL + TPA relative to KL alone. ($n = 3$ technical replicates, data are presented as mean values ± standard deviation). **e** qRT-PCR showing gene upregulation by TPA (1 hour) in combination with CDK9 inhibitors, flavopiridol (Flav) or NVP2, relative to DMSO control ($n = 3$ technical replicates, data are presented as mean values ± standard deviation). **f** Venn diagram (left) and heatmap (right) comparing 3-hour KL RNA-seq and 1-hour TPA RNA-seq differentially expressed (fold change ≥2, $P < 0.05$, two-tailed, Wald test) genes (Fisher's exact test, two-tail, $P = 2 \times 10^{-5}$). Source data are provided as a Source Data file.

nuclear proteins. In particular, HEXIM1 is among the established PKC substrates, and PKC-mediated HEXIM1 phosphorylation suppresses the formation of inactive PTEFb[30,31]. In this study, we identified a strong overlap between genes significantly changed with 1 hour of PKC induction and 3 hours of KL treatment. We established that the gene induction within 1 hour of PKC activation is dependent on CDK9 activity. We further demonstrated that PKC induction and KL treatment have minimal additive effects in activating differentiation-activating transcription factors, indicating that they impact the same pathway. Using co-immunoprecipitation, we identified that TPA or KL treatment was sufficient to disrupt the interaction between CDK9 and HEXIM1. Thus, these data link the earliest events of PKC signaling to the dissociation of inactive CDK9 from the AFF1-HEXIM1 repressive complex. Downstream to this dissociation, BRD4 facilitates the activation of differentiation activators, but not the activation of the rapid-response genes. A recent study also showed that BRD4 was not required for rapid-response to heat shock[49], suggesting other mechanisms are in place to directly engage the active CDK9 for the rapid-response genes.

Taken together, this study identified early molecular events underlying the initiation of the keratinocyte-differentiation cascade.

We identified rapid-response genes directly suppressed by AFF1-HEXIM1 in the progenitor state, through tethering CDK9 in an inactive state. Our findings contribute to a foundation for future studies examining CDK9 regulators and Pol II dynamics in somatic tissue homeostasis. Building on these findings highlighting the central roles of CDK9 activity switch, it will also be interesting to explore the chromatin binding and gene regulatory roles of the critical pause regulators NELF and DSIF in epidermal differentiation in the future.

## Method

This research complies with all relevant ethical regulations. This research was reviewed by Northwestern University Institutional Review Board (IRB) and assigned a determination of Not Human Research. The surgically discarded foreskin was obtained from Northwestern Skin Biology & Diseases Resource-Based Center and used to isolate primary human keratinocytes. This tissue was de-identified and considered discarded material according to IRB policy and thus, did not require patient consent. Northwestern University Institutional Review Board approved the protocol for tissue collection (IRB #STU00009443). All genomic sequencing data were generated using pooled keratinocytes isolated from at least 3-6 de-identified

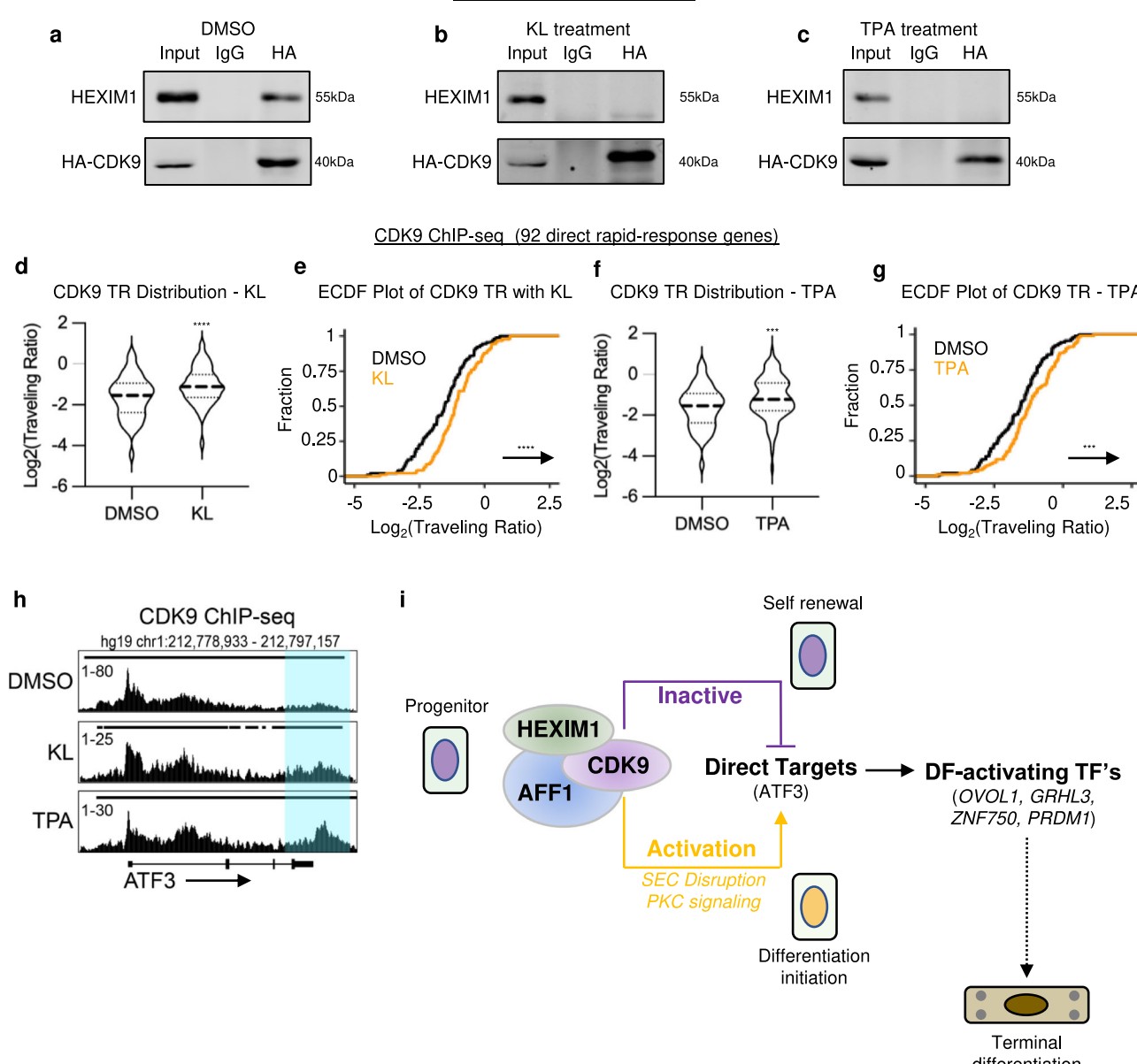

**Fig. 8 | CDK9 is dissociated from HEXIM1 upon KL or TPA treatment.**
**a**–**c** Keratinocytes expressing HA-CDK9 were treated with DMSO control, KL, or TPA. Immunoprecipitation using the HA antibody was performed from the lysate of these keratinocytes, followed by western blot probing for HA and HEXIM1. **d**–**g** Violin and ECDF plots showing HA-CDK9 traveling ratio (TR) with KL or TPA relative to DMSO control at 92, direct, rapid-response genes (****$P < 0.001$ ***$P < 0.01$, **d**, **e** KL $P = 0.0002$, **f**, **g** $P = 0.0041$, two-tailed, unpaired $t$ test). A higher travel ratio indicates high proportion of CDK9 binding in the gene body. **h** Genome browser tracks comparing HA-CDK9 ChIP-seq enrichment at ATF3 among the

DMSO control, KL treatment, or TPA treatment. **i** Illustration of the working model. In the progenitor state, AFF1 and HEXIM1 cooperatively hold CDK9 in an inactive state. With SEC disruption (KL) or PKC signaling (TPA), CDK9 from SEC-AFF1 is rapidly released into an active form. This rapid switch allows the activation of SEC-direct-target genes such as ATF3, which can further promote the expression of differentiation-activating transcription factors (GRHL3, PRDM1, ZNF750, and OVOL1, etc) to advance the terminal differentiation process. Source data are provided as a Source Data file.

donors, and because the sequencing data only covers a very small proportion of the genome, it is impossible to associate the genomic data with any distinct individuals.

## Cell culture
At least 3–6 de-identified donor keratinocytes were mixed and cultured together using a 1:1 mixture of Keratinocyte-SFM (Life Technologies #17005-142) and 50% Medium 154 (Life Technologies #M-154-500). To induce differentiation, keratinocytes were seeded at 100% confluency and treated with 1.2 mM CaCl$_2$ for four days. Phoenix and HEK293T cells were cultured in DMEM (Gibco) with 10% fetal bovine serum (HyClone). Devitalized human dermis for organotypic cultures was obtained from cadaver samples and was donated by the New York Firefighters Skin Bank.

## Plasmid construction
shRNA sequences were designed using ThermoFisher BOCK-iT™ RNAi Designer for shRNA's. Constructs were annealed and cloned into linearized pLKO.1 puro plasmid (Addgene #8453). The oligonucleotide sequences for AFF1 shRNA's were TAGGTTGGGAAAGCCGAAATA[18] and GCCTCAAGTGAAGTTTGACAA[50]. The oligonucleotide sequences for AFF4 shRNA's were GCACCAGTCTAAATCTATGTT[18] and GCACGAC

CGTGAGTCATATAA[18]. The oligonucleotide sequences for HEXIM1 were GCAGGAGCTCATCAAGGAGTA and GGAGTACCTGGAACTGGA GAA. For ATF3, DUSP1, and RND3 overexpression, ATF3 isoform 1 was amplified from differentiated keratinocyte cDNA by PCR. DUSP1 (Addgene #70317) and RND3 (Addgene #23229) were amplified by PCR. Constructs were then cloned into a Dox-inducible pTRIPZ (Dharmacon). The vector was modified by inserting a P2A cleavage sequence upstream to the RFP reporter. Cells were treated with 1.5 μg/mL of doxycycline to induce overexpression.

### Gene transfer and expression in keratinocytes
Phoenix and HEK293T cells were transfected using Turbofect or Lipfectamine 3000 based on manufacturer instructions. The virus was collected and filtered 48 and 72 hours after transfection. For infection, the virus was added to keratinocytes. Plates were centrifuged at $207 \times g$ for 1 hour at 32 °C, with polybrene (20 μg/mL). Puromycin (2 μg/mL) was added to cells 24 hours after infection and removed 48 hours after being added.

### Drug treatments
KL1 and KL2 peptidomimetic inhibitors[18] were resuspended in DMSO to 20 mM stock concentration. Cells were treated with 20 μM KL1 for both 3-hour and 24-hour treatments; however, due to the potency of KL2, cells were treated with 20 μM for 3-hour and 10 μM for 24-hour treatments. TPA treatments were done for 1 or 3 hours using 50 μg/mL TPA. CDK9 inhibition experiments were done with 800 ng/mL flavopiridol or 500 nM NVP2. PKC inhibition experiments were done with 1 μM Bisindolylmaleimide I (Abcam) or 1 μM Gouml6983 (Abcam).

### Immunofluorescence staining
Keratinocytes were seeded onto coverslips and fixed with 10% formalin. Fixed cells were permeabilized with PBST (0.4% Triton X-100) and subsequently blocked with 2.5% goat serum in PBST. Blocked coverslips were incubated with primary antibody overnight at 4 °C and then at room temperature for 1 hour with secondary antibody goat anti-mouse (Thermo Fisher, #A11005, 1:400) and/or goat anti-rabbit (Thermo Fisher, #A11034, 1:400). Nuclei were stained with NucBlue Fixed Cell Stain (Invitrogen). Imaging was done using an EVOS FL Auto 2 fluorescent microscope (Thermo Fisher). Mouse monoclonal anti-Ki67 (Santa Cruz Biotechnology, sc-23900, 1:50) and rabbit anti-p21 (12D1) (Cell signaling, 2947, 1:200) antibodies were used.

### Co-immunoprecipitation
About ten million keratinocytes were used per co-immunoprecipitation experiment. After cell trypsinization, nuclei were extracted in hypotonic buffer (10 mM Hepes pH 7.4, 1.5 mM MgCl$_2$, 10 mM KCL, 0.2% NP-40, 1× protease inhibitor EDTA free (Roche)). Nuclei were lysed in IP buffer (1 mM MgCl$_2$, 150 mM NaCl, 50 mM Tris pH 7.5, 5% glycerol, 1% NP-40, 1× protease inhibitor EDTA free) and sheared with a 27.5-gauge needle. Nuclei extraction was performed by incubating on ice for 30 minutes and subsequently spinning down the insoluble components at 21,130 g at 4 °C for 10 minutes. The supernatant was saved and used for co-immunoprecipitation.

For co-immunoprecipitation, 25 μL Dynabeads Protein G magnetic beads (Thermo Fisher) per condition were washed with PBST (0.002% Tween-20) three times. Beads were then incubated with 3 μg mouse IgG (G3A1) (cell signaling, 5415) or 3 μg mouse HA (66006-2-Ig, Proteintech) antibody for 10 minutes at room temperature and 2 hours at 4 °C on a tube rotator. The nuclear lysate was then added to beads overnight at 4 °C, rotating. Lysate for each condition was split evenly between IGG control and HA beads with a small aliquot saved for input loading control. Following overnight incubation, beads were washed three times with wash buffer (150 mM NaCl, 50 mM Tris pH 7.5, 0.05% NP-40, 1× protease inhibitor EDTA free). The sample was eluted in lysis buffer with BME and loading dye, shaking at 95 °C for 10 mins.

### Western blotting
In all, 15 μg of protein was loaded into each well of an SDS-PAGE gel and transferred onto a PVDF membrane. Blots were blocked at room temperature for 1 hour in Odyssey Blocker PBS (Li-COR) + 2.5% BSA. Blots were incubated with primary antibody overnight at 4 °C and with secondary antibody (goat anti-mouse, Li-COR, 926-68020, 1:20,000 and/or goat anti-rabbit, Li-COR, 926-32211, 1:20,000) for 1 hour at room temperature. Li-COR Odyssey Clx imaging system was used to image blots. Image Studio Software version 5.2 (Li-COR) was used for blot analysis. Antibodies used for western blotting included mouse monoclonal anti-Lamin A/C (E1) (Santa Cruz Biotechnology, sc-376248, 1:1000), rabbit polyclonal anti-AFF1 (Bethyl, A-3020344A-T, 1:500), rabbit polyclonal anti-AFF4 (Abclonal, A4644, 1:1000), rabbit polyclonal anti-HEXIM1 (Bethyl, A303-112A-T, 1:1000), rabbit monoclonal anti-ATF-3 (E9J4N) (Cell Signaling, 18665, 1:1000), rabbit monoclonal anti-DUSP1 (E8L7D) (Cell Signaling, 48625, 1:1000), mouse monoclonal anti-RhoE (RND3) (Cell Signaling, 3664, 1:300), Rabbit anti-PhosphoThr186 CDK9 (Cell Signaling, 2549, 1:1000), Rabbit anti-CDK9 (C12F7) (Cell Signaling, 2316, 1:1000), Rabbit anti-HA (C29F4) (Cell Signaling, 3724, 1:1000), Rabbit anti-DNMT1 (D63A6) (Cell Signaling, 5032, 1:1000), and Rabbit anti-MYC (D84C12) (Cell Signaling, 5605, 1:1000).

### Organotypic culture tissue regeneration
For progenitor competition assay, 0.5 million GFP-expressing control keratinocytes were mixed with 0.5 million DsRed-expressing control or knockdown keratinocytes. Cells were seeded onto devitalized human dermis and regenerated for 6–7 days. The tissue was then fixed by formalin and embedded with OCT. Frozen sections were stained briefly with DAPI and imaged for green and red fluorescence. For standard epidermal tissue regeneration with single-source keratinocytes, one million keratinocytes were seeded onto devitalized human dermis and regenerated for 6 days. Tissue was embedded with OCT. Cryosections of tissue were fixed before staining.

For H&E staining, the cryosections of tissue were fixed for 15 minutes with 10% formalin at room temperature. Following two 5-minute washes, the slides were incubated with Hematoxylin Gill III (Sigma) for 2 minutes, followed by an immediate wash with running tap water for 30 seconds. These slides were then placed in 70% ethanol for 30 seconds and 95% ethanol for 30 seconds. Each slide was incubated with 500 μL 0.5% Eosin for 30 seconds, followed by washes using ethanol and xylene. The slides were mounted using Xyl cytoseal before imaging.

### Colony formation assay
Mouse fibroblast 3T3 cells were treated in serum-free DMEM with mitomycin C (15 μg/mL) for 2 hours and subsequently seeded to a 6-well plate with ~8 × 10$^5$ cells per well. The following day, DMEM media was replaced with FAD 1 hour before the addition of keratinocytes. 800 keratinocytes were added per well. FAD media was replaced every two days until completion on the tenth day. 3T3 feeder cells were washed away with PBS, and keratinocytes were fixed with 1:1 acetone methanol for 5 minutes. Once dry, cells were stained with crystal violet. Quantification was completed by counting colonies >1 mm$^2$ in each technical replicate and comparing across different conditions.

### Quantitative real-time PCR
RNA extraction was performed with Quick-RNA™ MiniPrep (Zymo Research). cDNA synthesis was completed using the SuperScript VILO cDNA synthesis kit (Invitrogen). qPCR was run using the PowerUp SYBR Green Master Mix (Thermo Fisher) or the EvaGreen

Bullseye qPCR Master Mix (MidSci). Oligonucleotides used are included in Supplementary Data 8.

## RNA-seq

Quick-RNA MiniPrep Kit (Zymo Research) was used to extract RNA. Libraries for RNA-seq were prepared with NEBNext Ultra Directional RNA Library Prep Kit for Illumina (New England BioLabs) using Poly(A) mRNA Magnetic Isolation. 50-base-pair single-end reads were sequenced using Illumina HiSeq 4000 at Northwestern University NUSeq Core facility.

## ChIP-seq

Nuclei were extracted from keratinocytes and lysed. Lysate was sonicated using Bioruptor Pico (Diagenode) in cycles of 30 seconds on 30 seconds off for a total of 50–60 cycles. Immunoprecipitation was performed using Dynabeads Protein G (Thermo Fisher). Libraries were prepared using NEBNext Ultra II DNA Library Prep Kit for Illumina. Antibodies used for ChIP-seq were rabbit anti-RNA Pol II (D8L4Y) (Cell Signaling, 14958, 1.5 μL conjugated to 10 μL beads in 250 μL total volume), mouse monoclonal HA antibody(Proteintech, 66006, 6 μL conjugated to 10 μL beads in 250 μL total volume) for HA-CDK9, rabbit polyclonal anti-HEXIM1 (Abcam, ab25388, 3 μL conjugated to 10 μL beads in 250 μL total volume), and rabbit polyclonal AFF1[17] acquired from Dr. Ali Shilatifard's lab (5 μL conjugated to 10 μL beads in 250 μL total volume).

## Nascent RNA-seq

Nascent RNA was isolated using Click-It Nascent RNA Capture Kit (Thermo Fisher, C10365). Cells were labeled with 0.5 mM EU treatment for 1 hour. Cells were then collected in RNA lysis buffer and RNA was extracted with Quick-RNATM MiniPrep (Zymo Research). "Click-It" reaction was performed with 1–5 μg RNA. After this reaction, RNA was purified with RNA XP beads (Beckman Coulter). rRNA was depleted using rRNA depletion kit (NEB E7400s). Labeled RNA was then pulled down and converted to cDNA with Dynabeads MyOne Streptavidin T1 beads according to Click-It protocol. The resulting cDNA was purified with CleanNGS DNA binding beads. Libraries were prepared with Illumina DNA E7805 kit following protocol for inputs <100 ng.

## Statistical information

### qRT-PCR expression analysis

Each sample was run in technical triplicates using 18 S ribosomal RNA for normalization. This was done using QuantStudio Design & Analysis Software 1.3.1 (Thermo Fisher). A two-tailed, unpaired $t$ test was used for statistical analysis in GraphPad Prism version 7. Data are represented as mean plus or minus standard deviation. Statistical details are included in figure legends. Oligos used for qRT-PCR are included in Supplementary Data 8.

### Immunofluorescent staining analysis

At least 15 images were taken per condition using EVOS FL Auto 2 Imaging System Software Revision 2.0.1732.0 (Thermo Fisher). With Python 3[51], Scikit-image[52] was used to label nuclei and measure the total fluorescent intensity of Ki67 or p21 within each labeled nucleus. The mean fluorescent intensity for each image was calculated. The mean fluorescent intensity of each image was then made relative to the mean intensity of control images and plotted in GraphPad Prism version 7. An unpaired $t$ test was performed in GraphPad Prism for statistical analysis. This was completed for a total of three, consistent biological replicates; one representative biological replicate is shown.

### Progenitor competition assay quantification

Three biological replicates were performed; one representative replicate is shown. At least 10 images were taken of each organotypic culture. Fiji[53] version 2.1.0 was used to quantify total red and green fluorescent intensity in each image. The ratio of red to green fluorescence was calculated for each image. Values were plotted in GraphPad Prism. A Welch $t$ test was performed in RStudio[54] version 3.6.2 for statistical analysis.

### Organotypic culture thickness quantification

Ten images were taken for each tissue section. Four different regions were measured in the same image using ImageJ version 2.1.0 and an average was recorded. The average thickness of these 10 images was used to represent the thickness of one tissue. Three tissues were regenerated per condition.

### ChIP-seq data analysis

ChIP-seq reads were aligned using Bowtie2[55]. Peaks were called using MACS2[56]. BEDtools[57] was used to compare peaks between data sets. Average profile plots and heatmaps were generated using deepTools[58]. Box plots and violin plots were generated using GraphPad Prism version 7. Statistical analysis for box and violin plots was performed in GraphPad Prism version 7 with unpaired $t$ tests. The empirical Cumulative Density Function (ECDF) plot was generated in RStudio[54] using ggplot2[59]. A Welch $t$ test was performed in RStudio[54] for statistical analysis of ECDF plots. EdgeR[60] was used to generate heatmaps showing differential pausing analysis of Pol II ChIP with TPA and KL treatment.

### RNA-seq data analysis

RNA-seq reads were aligned using HISAT2[61]. SAMtools was used for sorting files[62]. Counts tables were built with HTSeq[63]. Differential expression analysis was done using DESeq2[64]. Heatmaps were made with ggplot2[59] in RStudio[54]. Gene Ontology analysis was completed using DAVID[65].

### Reporting summary

Further information on research design is available in the Nature Research Reporting Summary linked to this article.

## Data availability

RNA-seq and ChIP-seq data have been deposited at GEO (GSE#182959) and are publicly available. Any additional information required to reanalyze the data reported in this paper is available from the lead contact upon request. Source data are provided with this paper.

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

## Acknowledgements

This work is supported by an NIH K99/R00 Award (R00AR065480) to X.B., an NIH R01 (AR075015) to X.B., a Research Scholar Grant (RSG-21-018-01-DDC) from the American Cancer Society to X.B., the Searle Leadership Fund to X.B., the Northwestern Skin Disease Research Center Pilot & Feasibility Award to X.B., the Basic Insights Award from Northwestern Cancer Center to X.B., a NUCATS Pilot Award to X.B., an NIH CMBD training grant (T32GM008061) to S.M.L. as a trainee, a Northwestern Presidential Fellowship to S.M.L., Northwestern Summer Undergraduate Research Grants (L.A.B., M.O.B., M.P.M., D.B.L., D.R.), Northwestern Academic Year Undergraduate Research Grants (M.O.B., M.P.M., D.B.L.), and Weinberg College Summer Research Grants (D.B.L., D.R., L.A.B., M.O.B.). We appreciate the support from the Skin Biology and Diseases Resource-based Center (SBDRC, P30AR075049) for providing tissues and culture media for this study, and the NUseq facility for providing next-generation sequencing service for this study. We thank Kaiwei Liang, Yuki Aoi, Edwin R. Smith, and Ali Shilatifard for sharing the KL1 and KL2 inhibitors, the AFF1/4 antibodies, as well as a subset of shRNAs used in this study. These reagents were instrumental in making this project possible. We also appreciate the suggestions from the Shilatifard Lab on this project.

## Author contributions

S.M.L. completed most of the experiments and data analysis. D.B.L. performed tissue sectioning, staining, and quantification. M.O.B., D.R., and M.P.M. contributed to cloning, inhibitor treatments, and qRT-PCR. J.K. and A.E.N. assisted with RNA-seq and ChIP-seq library construction. L.A.B. performed several western blots. P.J.H. assisted with ChIP-seq analysis. X.B. and S.M.L. designed this study and wrote this manuscript in consultation with other authors, and in discussions with other colleagues at Northwestern University.

## Competing interests

The authors declare no competing interests.
