## [Peer Review File · Nature Communications]

CDK9 activity switch associated with AFF1 and HEXIM1
controls differentiation initiation from epidermal progenitorsREVIEWER COMMENTS

Reviewer #1 (Remarks to the Author):

This manuscript describes a role for SEC in promoting progenitor self-renewal by directly repressing genes involved in epidermal differentiation. The authors have used a relevant primary cell culture model to investigate epidermal differentiation. They induced differentiation by CaCl₂ rather than confluency.

Overall, this is a promising but underdeveloped paper that I think would benefit from additional clarity on the mechanism of action and readout assays. Most concerning, the authors rely primarily on in vitro primary cell cultures and drug treatment of cells. They did not perform in vivo validation or functionality assays to validate their results on epidermal differentiation, with no experiment on self-renewal except the organotypic culture in Fig.3c using AFF1-KD cells. Additional assays on differentiation and barrier function are desirable to back up the gene expression data.

Major comments:

1. Rapid response genes are shown to be regulated at the mRNA level. Can the authors show protein expression and evidence of their subsequent effect on differentiation?

2. Functionality assays are absent. Proof-of-concept assays should be performed. Per example the authors do not show the impact of loss of differentiation on barrier function and the effect of premature differentiation on epidermal function. In fact, it seems contradictory in the progenitor competition assays that the tissue is not thin(ner) if the cells enter a premature post-mitotic differentiation? Can the authors clarify this data and provide functional in vivo assays?

3. The authors claim that progenitors either self-renew or differentiate however the results do not show self-renewal readout. It was difficult to follow at times what is the expected effect on self-renewal potentials. Can the authors sort for specific cell populations (progenitors vs differentiated) that can be used for ChIP-seq and/or RNA-seq.

Minor comments:

1. Can the authors instead of KL treatment generate SEC-knockout (or knockdown) cells and treat them with CaCl₂ and assess for early differentiation?

2. Different layers of the epidermis express different levels of differentiation genes. Does the AFF1-SEC-pTEFb repressive complex persist throughout the differentiated and terminally differentiated layers of the epidermis?

3. I thought the link to PKC activation by TPA and the induction of differentiation would benefit from additional details on PKC impact (phosphorylation) on downstream substrates. How can the authors explain the role of transient TPA induction of early differentiation versus TPA-induced inflammation and hyperproliferation of the epidermis?

Reviewer #2 (Remarks to the Author):

A molecular switch of SEC-PTEFb activity controls progenitor self-renewal versus differentiation-initiation

Sarah M. Lloyd, Mari O. Brady, Deborah Rodriguez, Daniel B. Leon, Madison P. McReynolds, Junghun

Kweon, Amy E. Neely, and Xiaomin Bao

Previous studies have shown that the RNAP II pausing at the promoter-proximal regions is the significant rate-limiting step to regulate the mammalian gene expression. Lloyd et al showed that the keratinocyte differentiation is primarily involved in the RNAP II release from the TSS using various techniques. The gene expression pattern following differentiation is similar to that of P-TEFb dissociation by KL and siAFF1KD, and siHEXIM1KD. The upregulation of the differentiation-associated gene results from the expression of rapid response genes, such as ATF3. In addition, they showed how the keratinocyte differentiation initiate using PKC signaling, which is previously revealed to be involved in promoting keratinocyte differentiation. Inhibition of CDK9 kinase activity suppresses the PKC signaling, suggesting the PKC signaling is also implicated in the P-TEFb dissociation. The data of this manuscript show a high correlation between them. However, several questions remain to reviewer, such as "How could P-TEFb dissociation release the paused RNAP II and activate the several genes?" Despite the highly correlated data, the data description is insufficient to understand the mechanism.

The RNAP II promoter-proximal pausing release mainly involves pausing factors, DSIF and NELF, in higher eukaryotes. The P-TEFb phosphorylates these factors to release the RNAP II and maintain the productive elongation rate. Therefore, DSIF and NELF should be considered in the release factors of RNAP II pausing. In addition, P-TEFb is believed to be the most active when associated with the SEC. Treatment of KL, siAFF1/4KD, and HEXIM1KD will be involved in the dissociation of P-TEFb from the SEC. I wonder how the dissociated P-TEFb could phosphorylate the pausing factors and RNAP II CTD to release pausing to activate differentiation-associated genes. Indeed, Liang et al have shown that KL treatment induces the pol II pausing at the TSS site and reduces the pol II elongation rate.

As P-TEFb is involved in RNA production, the spike-in normalization should be required in RNA-seq. However, I could not find the spike-in normalization. If the global RNA production decreased upon SEC dissociation, all of the data related to RNA (RT-qPCR, RNA-seq) could be misinterpreted because control RNA could be affected. To convince me of the unimaginable data, more exquisite data normalization is required.

Unfortunately, the logical flow in this manuscript is unsatisfactory. It would be worthwhile to suggest the possible mechanism more reasonably. Especially, I wonder how the inactive form of P-TEFb can be activated upon dissociation from AFF1. After sufficient discussion, the authors could change model figure (4m, 7g) to more accessible.

Specific Comments:

1. Line 42~43. In your manuscript, SEC suppresses the genes related to development and differentiation. It is not appropriate to suggest 'promoting' the progenitor self-renewal. If promoting self-renewal, the gene related to self-renewal should be down-regulated upon KL treatment or AFF1 knock-down.
2. Line 138. The gene groups of cluster II show the up-regulation upon differentiation, although their genome-wide pol II occupancy decreased. What do you think of the reduced pol II level on gene body regions inducing the up-regulation of those genes?
3. Line 135~138, Figure 1.c, e. I think the peak pattern of the example does not match the metaplot.
4. Line 143~145. As you used the pausing index to explain the mechanism for regulation of the genes, the high pausing index shows the correlation of high gene expression. In figure a, however, the gene expression on cluster I seems to show no gene expression despite the high pol II pausing at TSS in an undifferentiated state.

5. Figure 1.i. P-TEFb phosphorylates DSIF and NELF to release the pol II pause. It means that the pause release and gene activation without P-TEFb is not persuasive. According to Liang et al (2018, Cell), the KL treatment causes degradation of AFF1/4 and the decrease of pol II elongation rate. However, 3-hour KL treatment makes several genes upregulated. How can you explain these genes upregulated?
6. Figure 1.k, l, m. In figure 2, the RNA-seq DEG heatmaps upon 24-hour KL treatment were shown. I'm wondering their gene groups show similar patterns with those upon 3-hour treatment. It will be good if those data upon 24-hour KL are also shown.
7. Line 203, supplementary figure 2a, b. The western blot band is not clear. I suggest the western data should be more obvious, like figure 4c.
8. Supplementary figure 2d. Figure 2d is written two times. AFF1sh-1/2 might be figure 2c.
9. Line 214~215. Figure 3c. What do you think of the reason for the accumulation of the regenerated red tissue? This difference between AFF1KD and AFF4KD is surprising, so I'm curious about the reason. In addition, AFF1sh-1/2 all shows this phenomenon?
10. Line 237~240. Figure 3k~m. This sentence is ambiguous to understand. The reason that AFF1 knock-down and KL treatment would show 'minimal' effect should be explained more. KL treatment causes the dissociation of P-TEFb, so AFF1 knock-down of the KL treatment sample will not give more changes than only the KL treatment sample. If that is the intention, I recommend the summation of the figure (k and l) and (k and m).
11. Figure 3k~m. Different from the previous figure 2f~i, which type of KL was used?
12. Figure 4g~i. Why is only KL2 used in this RT-PCR?
13. Line 265~266. Figure 4g~i. It is not clear. Through figure 2, KL treatment itself seems to induce differentiation. Why do you mention 'unable to further induce differentiation?' In this sentence, what is the control, either control KD or HEXIM1KD+DMSO? As for question 10, I recommend the summation of the figure (g and h) and (g and i).
In addition, after KL treatment, P-TEFb can be dissociated from AFF1/4. However, how can prove that KL treatment causes the dissociation of P-TEFb from HEXIM1? Which explanation is plausible, HEXIM1 is also dissociated from P-TEFb, or P-TEFb can be activated despite the binding of HEXIM1 without AFF1? In figure 4a-b, P-TEFb can be activated after the dissociation from AFF1. If the control and KL2 treatment in the HEXIM1KD does not show differences, it seems that HEXIM1 itself cannot block the activation of P-TEFb.
14. Figure 5e. I could understand the reason that RNA-seq upon 3hr KL treatment. But I wonder how many overlapping genes exist upon 24hr KL treatment.
15. Figure 5i~k. The ChIP-seq example on the region of the differentiation-activating TFs (GRHL3, OVOL1, PRDM1, ZNF750) is required.
16. Figure 5i~k. How is the distribution of RNAP II upon KL treatment, AFF1KD, or HEXIM1KD?
17. To make figure 3~4 more reliable, P-TEFb ChIP-seq upon AFF1KD and HEXIM1KD is required. The P-TEFb distribution will not be changed? If not so, how could the dissociation of P-TEFb from SEC induce the gene activation?
18. Figure 6. In the title of this manuscript, SEC-pTEFb itself seems to induce the keratinocyte

differentiation. However, the differentiation seems to result from an indirect effect through figure 6. In addition, the 3hr KL treatment is sufficient to induce the differentiation-activating TFs. Which genes are first activated after KL treatment, ATF3 or differentiation-activating TFs? Short treatment of KL should be required.

19. Line 330~332. Supplemental figure 5b. Although some articles suggest the calcium-mediated keratinocyte differentiation, no change of ATF3 gene in calcium insertion might implicate that ATF3 itself is not sufficient to induce the differentiation. What about the clonogenic assay upon calcium treatment?

20. Line 341~345. Figure 7a. If ATF3 directly induces the differentiation activating genes, ATF3 expression should be maintained while those genes are activated. Why does the expression of the ATF3 gene decrease after 3hr?

21. Figure 7c, d. What about the summation of the figure 7c and 7d? The graph of Figure 7d is too short of comparing.

22. Figure 7e. What if Cdk9 inhibition after KL, siAFF1, or siHEXIM1?

23. Line 429 ~ 430. It might be possible that the difference with the previous data related to pol II pausing might result from the use of primary cell. How about showing the difference between primary cell and immortalized cell upon KL or AFF1KD treatment?

24. Line 562. The exact number of cycles might be written.

Reviewer #3 (Remarks to the Author):

In this study, Llyod et al. carried out Pol II ChIP-seq and combined this analysis with their previously reported RNA-seq data to determine Pol II dynamics during KC differentiation. This analysis allowed them to identify a cluster of genes that were highly enriched for paused Pol II in undifferentiated KCs and robust Pol II elongation in differentiated KCs. Interestingly, disruption of PTEFb-SEC association for 3 hours results in upregulation of several Cluster I genes in UD KCs, and disruption for 24 hours leads to a larger gene expression change which includes upregulation of several genes that are expressed in DF KCs. Further studies revealed that loss of AFF1, and not AFF4, results in the upregulation of genes associated with epidermal differentiation. Additionally, HEXIM1 synergistically associates with AAF1-SEC to maintain target gene repression. The authors further show that the activity switch of SEC-PTEFb mediates the early events of PKC signaling which is important for keratinocyte differentiation.

Overall, this study has identified that the activity switch of SEC-PTEFb is responsible for regulating the earliest events that promote the fate-choice decisions between self-renewal and differentiation. Findings reported in this paper add to the knowledge of Pol II dynamics between UD and DF states of KC and will be of interest to researchers working in the broader community of gene regulation during differentiation. Although this paper does a thorough job in addressing the gene expression changes under the control of HEXIM-SEC-PTEFb mediated RNA Pol II pausing in maintaining progenitor fate, it fails to address some fundamental questions mentioned below. The following comments and concerns need to be addressed in this manuscript before it is considered for publication.

Major Comments:

1. In the introduction the authors mention that while the inactive form of PTEFb associates with a complex containing HEXIM1, the active form of PTEFb associates with SEC. But in this study, they conclude that HEXIM1 works synergistically with AFF1-SEC by maintaining CDK9 in an inactive state. Therefore, to fully support the model reported in Figure 4m, the authors need to conduct biochemical assays to show the physical interaction of these proteins in KCs. An IP with antibodies against CDK9 and pCDK9 and then followed by western blot on IP product to show an association of HEXIM1 and AFF1 with inactive pTEFb is absolutely needed. Moreover, is 7SK snRNA also part of this synergistic complex? Depletion of 7SK snRNA results in untethering of HEXIM to inactive PTEFb which allows for PTEFb phosphorylation and Pol II elongation. Please address.

2. The authors show that HEXIM1 and AFF1 specifically target "rapid response" genes in UD KCs to stall Pol II at the proximal promoter regions based on ChIP-seq results. However, the authors do not address what is unique about these genes that result in the association of HEXIM-inactive pTEFb with AFF1-containing SEC. Are there DNA sequences common between these differentiation-promoting genes that allow for this unique targeting in UD KCs? Are there epigenetic signatures unique to these genes?

3. Does PKC activation result in dissociation of HEXIM1 from these repressed genes allowing for gene expression during differentiation? To show a direct correlation of AFF1-PTEFb-HEXIM1 disruption upon PKC signaling, and Pol II elongation resulting in expression of differentiation promoting TFs, the authors need to conduct ChIP analysis of AFF1, HEXIM1, and Pol II on target genes in UD KCs treated with TPA. Also, an IP of CDK9 in KCs treated with TPA can also clarify if the AFF1-PTEFb-HEXIM1 is altered. Also, does PKC simply results in dissociation of HEXIM1, or its expression is diminished upon PKC signaling?

Other Comments:

1. Given that the authors see a clear distinction in Pol II occupancy at TSS and gene body in UD vs DF states of KC, how does the distribution of pSer5 vs pSer2 look like in these states? Pol II CTD accumulates pSer2 as it progresses through a gene body, therefore is this accumulation evident in the DF state of KC? Can the authors perform IF or Western Blot analysis with pSer2 and pSer5 in UD and DF to show if the phosphorylated state of Pol CTD residues mirrors the ChIP data? Alternatively, ChIP-qPCR using pSer2 and pSer5 antibodies on select Cluster I genes can also address this.

2. It looks like Cluster II genes have lower Pol II occupancy at TSS even in UD KC but accumulate Pol II elongation in DF KCs. Is this because chromatin dynamics (repressed chromatin domains) play a role in inhibiting Pol II access to Cluster II genes at UD state? Moreover, Cluster III genes are downregulated in DF KCs, indicating that they are actively transcribed in UD KCs. Therefore, it is important to include Pol II enrichment in gene bodies of Cluster III genes in Sup Fig 1e – which should be high in UD and low in DF compared to Cluster I.

3. The authors mention the number of shared genes between KL treated and DF KCs, but it will be helpful if they can also indicate how many of the shared genes that are upregulated in KL treated UD KCs are also expressed in DF KCs? Is there a statistical significance in this overlap? Maybe a GSEA analysis using DF gene set as the background will be helpful.

4. The authors at the end of Figure 2 conclude that "SEC-PTEFb disruption progressively upregulates terminal differentiation and diminishes proliferation", while the latter was shown by IF of Ki67 and cell numbers, the authors need to carry out IF of classical differentiation markers to show that UD cells when treated with KL take up on a differentiated fate.

5. The authors carried out PCA analysis to show KL treated UD KCs were similar to that of AFF1sh KCs, but they don't report the number of shared genes that are upregulated in both instances. The authors need to provide a Venn or a graph showing how many genes upregulated upon AFF1sh are also

upregulated in 24-hour KL treatment.

6. Fig. 3k-m is a bit confusing. AFF1 KD leads to significant upregulation of TFs, as shown in Fig. 2h, but in 3l and m in the DMSO panel, the upregulation of these TFs is not as apparent (y-scale on 2h was up to 20-30 while on 3l is below 5). Please address this discrepancy. Also, why would AFF1 KD mediated upregulation "mitigate" the effects of KL treatment, if they are both targeting the SEC-PTEFb association? I have the same concern with Fig 4f and 4h. Why doesn't the HEXIM KD without KL show the same level of upregulation as compared to shown in 4f? Please explain.

Point by Point Response to Reviewers

(Reviewers' original comments in Gray, responses in Black, new data in figures in Blue, information for reviewers in Brown. Reviewer Figures are included in pages 13-20 of this file)

Reviewer #1:

This manuscript describes a role for SEC in promoting progenitor self-renewal by directly repressing genes involved in epidermal differentiation. The authors have used a relevant primary cell culture model to investigate epidermal differentiation. They induced differentiation by CaCl₂ rather than confluency.

Overall, this is a promising but underdeveloped paper that I think would benefit from additional clarity on the mechanism of action and readout assays. Most concerning, the authors rely primarily on in vitro primary cell cultures and drug treatment of cells. They did not perform in vivo validation or functionality assays to validate their results on epidermal differentiation, with no experiment on self-renewal except the organotypic culture in Fig.3c using AFF1-KD cells. Additional assays on differentiation and barrier function are desirable to back up the gene expression data.

We appreciate the positive feedbacks from the reviewer that this is a “promising” paper. We would like to thank the reviewer for the helpful suggestions on improving “clarity on the mechanism of action” as well as including more “readout assays” and “functional assays”. We have incorporated these in the revision, with details included in the following paragraphs.

We apologize for the confusion regarding the differentiation methods used for different experiments. We have now clarified in the results/methods section that 1.2mM CaCl₂ **and** 100% confluency was used to induce differentiation in the context of RNA-seq and ChIP-seq in the “DF” conditions. The only one experiment performed with 1.2mM CaCl₂ but at low confluency was designed to determine whether increased extracellular calcium alone is sufficient to induce rapid upregulation of AFF1 direct targets (Supplementary Figure 7b). This experiment showed that unlike high CaCl₂ **and** 100% confluency for 48 (D2) or 96 (D4) hours (Supplementary Figure 7a), calcium for only 3 hours with sub-confluent keratinocytes did not drastically upregulate gene expression.

Major comments:

1. Rapid response genes are shown to be regulated at the mRNA level. Can the authors show protein expression and evidence of their subsequent effect on differentiation?

We have confirmed that the rapid-response genes (ATF3, DUSP1, and RND3) are upregulated at the protein levels within 3 hours of KL treatment (new data included as Supplementary Figure 5e-h).

In the context of their subsequent effect on differentiation, we identified that ATF3, but not DUSP1 or RND3, was sufficient to impair clonogenicity and to upregulate representative differentiation activators (Figure 6). We further confirmed that the upregulation of differentiation activators, such as PRDM1, was evident at the protein level with ATF3 overexpression (new data included as Reviewer Fig 1a,b). Using organotypic epidermal regeneration assay, we found that ATF3 overexpression led to tissue hypoplasia (new data included as Supplementary Figure 6a,b), indicating that ATF3 overexpression is sufficient to impair keratinocyte regenerative capacity.

2. Functionality assays are absent. Proof-of-concept assays should be performed. Per example the authors do not show the impact of loss of differentiation on barrier function and the effect of premature differentiation on epidermal function. In fact, it seems contradictory in the progenitor competition assays that the tissue is not thin(ner) if the cells enter a premature post-mitotic differentiation? Can the authors clarify this data and provide functional in vivo assays?

We appreciate these comments from the reviewer. To clarify and strengthen the findings from the competition assay, we measured and quantified the epidermal thickness. Also, since originally we performed three replicates using AFF1 sh1 and AFF4 sh1, we have now additionally performed three replicates of the competition assay with AFF1 sh2 and AFF4 sh2 (new data included as Supplementary Figure 3g,h). The reduction of epidermal thickness with AFF1/4 knockdown was minimal in all replicates of competition assays (new quantification included as Reviewer Figure 2a,b). This is likely due to control cells compensating and outcompeting knockdown cells. To further clarify the impact of knockdown on premature differentiation induced by knockdown, we regenerated epidermal tissue organotypically with knockdown cells alone. These showed a more drastic reduction in epidermal thickness, especially with AFF1 knockdown (new data and quantification

included as Supplementary Figure 3i,j), indicating a loss of the ability to regenerate a stratified tissue. We also tried *in vivo* competition assay, which involves grafting the regenerated epidermal tissue on the backs of immunodeficient mice. The results were consistent with the results from organotypic culture (new data included as Reviewer Figure 2c).

The reviewer brought up a very interesting question regarding the roles of AFF1 in epidermal terminal differentiation. The action of specific “transcription regulators” in progenitors state versus differentiated cells can be highly context dependent. The lineage-specific transcription factor p63 represses differentiation in the progenitor state, and activates differentiation in the terminal differentiation process. The BAF (SWI/SNF) remodeling complex is another example. BAF perturbation de-represses differentiation in the progenitors, but in the differentiated state BAF activity is essential for activating terminal differentiation markers such as Krt10 (Bao et al, 2013, Bao et al 2015). With AFF1 knockdown, but not AFF4 knockdown, we found that the activation of Krt10 in epidermal differentiation is drastically impaired (new data included as Reviewer Figure 2d,e). Thus, AFF1’s role in the context of differentiation activation state is different from its role in the progenitor state. Consistently, we were able to co-immunoprecipitate AFF1 and HEXIM1 in the progenitor-state keratinocytes, but this interaction was lost in differentiated keratinocytes (new data included as Figure 4m,n). Thus, AFF1’s gene regulatory roles in differentiated keratinocytes are likely to be very different from its action in the progenitor-state keratinocytes. We have also added these comments in the “discussion” section.

3. The authors claim that progenitors either self-renew or differentiate however the results do not show self-renewal readout. It was difficult to follow at times what is the expected effect on self-renewal potentials. Can the authors sort for specific cell populations (progenitors vs differentiated) that can be used for ChIP-seq and/or RNA-seq.

We apologize for the confusion and the lack of clarification in the previous version of the manuscript. Two functional assays that we used for assessing “self-renewal potentials” included clonogenicity assay and progenitor competition assay. In the revised manuscript, we have included a third approach to assess the thickness of epidermal tissue regenerated using just AFF1/4 knockdown keratinocytes, without mixing them together with control cells in the same tissue. We also included new data of epidermal regeneration using keratinocytes with ATF3 overexpression as compared to control (new data included as Supplementary Figure 3i,j, Supplementary Figure 6a,b). AFF1/4 knockdown or ATF3 overexpression abolished keratinocytes’ ability to regenerate full-thickness epidermal tissue. These results are consistent with clonogenicity and progenitor competition assays that we had included in the previous version.

To further clarify how AFF1 or AFF4 influences self-renewal in keratinocytes, we assessed the protein levels of representative “self-renewal” marker genes MYC and DNMT1. We found that AFF1/4 knockdown or KL treatment strongly reduced the protein levels of MYC and DNMT1 using western blotting (new data included as Supplementary Figure 3c-f, Supplementary Figure 2a,b). Based on our AFF1 ChIP-seq data, we identified direct AFF1 binding to the promoters of both MYC and DNMT1 in the progenitor-state keratinocytes (new data included as Supplementary Figure 5k). These data suggest that AFF1 also functions to sustain the expression of key self-renewal genes. Notably, the downregulation of these self-renewal genes is not part of the immediate responses. Downregulation of MYC and DNMT1 was not detectable within 3 hours of TPA treatment based on our mRNA-seq and nascent RNA-seq data. The downregulation was detectable, however, after 24-hour of KL treatment (new data included as Supplementary Figure 5m).

We appreciate the question of sorting specific cell populations for ChIP-seq and/or RNA-seq. This specific project is focused on characterizing rapid responses within as short as 1 hour, in a relatively homogenous cell population. This cell sorting approach will be more beneficial in other projects where the starting materials are more heterogenous and the experimental timing is less restrictive.

Minor comments:

1. Can the authors instead of KL treatment generate SEC-knockout (or knockdown) cells and treat them with CaCl₂ and assess for early differentiation?

We treated AFF1 knockdown cells with CaCl₂ and assessed early differentiation. We observed no drastic changes in gene expression (new data included as Reviewer Figure 3a,b).

2. Different layers of the epidermis express different levels of differentiation genes. Does the AFF1-SEC-

pTEFb repressive complex persist throughout the differentiated and terminally differentiated layers of the epidermis?

We appreciate this good question from the reviewer! To determine if AFF1-SEC-pTEFb repressive complex persists in differentiation, we performed co-immunoprecipitation experiments using undifferentiated or differentiated keratinocytes. We found that AFF1 binds to CDK9 in both conditions; however, AFF1 only interacts with HEXIM1 in undifferentiated keratinocytes (new data included as Figure 4m,n). These data demonstrate that the AFF1-SEC-pTEFb repressive complex does not persist in differentiation.

3. I thought the link to PKC activation by TPA and the induction of differentiation would benefit from additional details on PKC impact (phosphorylation) on downstream substrates. How can the authors explain the role of transient TPA induction of early differentiation versus TPA-induced inflammation and hyperproliferation of the epidermis?

We appreciate these questions from the reviewer. HEXIM1 has been reported as a substrate of PKC (Fujinaga et al., 2012). PKC activation leads to HEXIM1 phosphorylation and subsequent activation of CDK9. Consistent with HEXIM1 being a PKC substrate, we found that the induction of early differentiation by TPA is dependent on CDK9 activity (Figure 7e). We have clarified this in the manuscript.

TPA is used as a tumor-promoting agent in the classic DMBA-TPA approach to induce carcinogenesis in mouse epidermis. On the other hand, TPA and PKC activation has been reported to induce differentiation in human keratinocytes (Jerome-Morais et al., 2009; Tibudan et al., 2002). To address the differences, we first compared ATF3 protein sequence between human vs mouse, and we identified several amino acid substitutions. We then overexpressed the mouse version of ATF3 in human keratinocytes. We found that the expression of mouse ATF3 also induced the expression of differentiation marker genes such as GRHL3, OVOL1 and PRDM1 (new data included as Reviewer Figure 4a). Therefore, the induction of differentiation genes by TPA is not due to the differences of amino acid composition between mouse versus human.

In addition to HEXIM1, PKC has a broad range of substrates. When keratinocytes were treated with TPA for 1 hour, only 169 genes were significantly changed (fold change >2, p<0.05) as compared to control (DMSO); however, with 3 hours of TPA treatment, the number of significantly changed genes increased to 855 and only 14% of them overlap with 3 hours of KL treatment (new data included as a reviewer Figure 4b), suggesting that additional PKC substrates or even indirect substrates could contribute to the profound changes at later time points.

Digging deeper into the literature, we noticed an interesting paper published back in 1982 (Yuspa et al., Cancer Research). In this paper, the authors cultured mouse epidermal basal cells and treated them with TPA. Within 12 hours of TPA treatment, epidermal transglutaminase activity was found to increase 2-4 times, suggesting increased differentiation (in agreement with what we observed in this study). However, when these mouse epidermal basal cells were treated with TPA for a second time, proliferation increased. When we treated primary human keratinocytes with TPA, we observed that CDK9 was dissociated from HEXIM1 (new data included as Fig. 8a,b). If a second dose of TPA is added to keratinocytes while CDK9 is already dissociated from HEXIM1, this would not lead to another wave of differentiation induction. In terms of the mechanisms related to how TPA stimulates proliferation, a different paper (Su et al., PNAS, 2018) reported that TPA can stabilize a different kinase --- CK1e, which can further lead to activation of the wnt pathway. Taken together, PKC activation by TPA is connect to differentiation activation; however longer-term repeated TPA treatment are associated with the “off-target” effects of this chemical, including the stabilization of CK1e and the activation of the wnt pathway to stimulate proliferation.

Reviewer #2:

A molecular switch of SEC-PTEFb activity controls progenitor self-renewal versus differentiation-initiation
Sarah M. Lloyd, Mari O. Brady, Deborah Rodriguez, Daniel B. Leon, Madison P. McReynolds, Junghun Kweon, Amy E. Neely, and Xiaomin Bao

Previous studies have shown that the RNAP II pausing at the promoter-proximal regions is the significant rate-limiting step to regulate the mammalian gene expression. Lloyd et al showed that the keratinocyte differentiation is primarily involved in the RNAP II release from the TSS using various techniques. The gene expression pattern following differentiation is similar to that of P-TEFb dissociation by KL and siAFF1KD, and

siHEXIM1KD. The upregulation of the differentiation-associated gene results from the expression of rapid response genes, such as ATF3. In addition, they showed how the keratinocyte differentiation initiate using PKC signaling, which is previously revealed to be involved in promoting keratinocyte differentiation. Inhibition of CDK9 kinase activity suppresses the PKC signaling, suggesting the PKC signaling is also implicated in the P-TEFb dissociation.

The data of this manuscript show a high correlation between them. However, several questions remain to reviewer, such as “How could P-TEFb dissociation release the paused RNAP II and activate the several genes?” Despite the highly correlated data, the data description is insufficient to understand the mechanism. The RNAP II promoter-proximal pausing release mainly involves pausing factors, DSIF and NELF, in higher eukaryotes. The P-TEFb phosphorylates these factors to release the RNAP II and maintain the productive elongation rate. Therefore, DSIF and NELF should be considered in the release factors of RNAP II pausing. In addition, P-TEFb is believed to be the most active when associated with the SEC. Treatment of KL, siAFF1/4KD, and HEXIM1KD will be involved in the dissociation of P-TEFb from the SEC. I wonder how the dissociated P-TEFb could phosphorylate the pausing factors and RNAP II CTD to release pausing to activate differentiation-associated genes. Indeed, Liang et al have shown that KL treatment induces the pol II pausing at the TSS site and reduces the pol II elongation rate.

As P-TEFb is involved in RNA production, the spike-in normalization should be required in RNA-seq. However, I could not find the spike-in normalization. If the global RNA production decreased upon SEC dissociation, all of the data related to RNA (RT-qPCR, RNA-seq) could be misinterpreted because control RNA could be affected. To convince me of the unimaginable data, more exquisite data normalization is required.

We thank the reviewer for pointing out this important issue. We quantified nascent RNA production comparing SEC perturbation by KL versus control, and found no significant changes (new data included as a reviewer figure 5). We also compared our AFF1 ChIP-seq and HEXIM1 ChIP-seq with CDK9 ChIP-seq data. AFF1 and HEXIM1 ChIP-seq peaks only overlap with a small subset of total CDK9 peaks (new data included as supplementary Figure 5a). SEC dissociation does not lead to global RNA production changes. In addition, we have attempted to use spike-in control according to the cell number count. However with the current technical capacity, we found that the spike-in introduced high level of variations even within technical replicates, likely due to the limitation of manual cell counting and pipetting errors. Thus, using sequencing counts to normalize the RNA-seq data represents a better unbiased way for data analysis in this context, when only a small subset of genes were altered with perturbation.

Unfortunately, the logical flow in this manuscript is unsatisfactory. It would be worthwhile to suggest the possible mechanism more reasonably. Especially, I wonder how the inactive form of P-TEFb can be activated upon dissociation from AFF1. After sufficient discussion, the authors could change model figure (4m, 7g) to more accessible.

We appreciate the suggestions from the reviewer. We have added CDK9 ChIP-seq data to Figures 5 and 8 (new data included as Supplementary figure 5a-c, Figure 8d-h) and co-immunoprecipitation experiments to Figures 4 and 8 (new data included as Figure 4m,n, Figure 8a-c). We have also updated the models in Figure 5 and Figure 8, with the new data we obtained during the revision.

Specific Comments:

1. Line 42~43. In your manuscript, SEC suppresses the genes related to development and differentiation. It is not appropriate to suggest ‘promoting’ the progenitor self-renewal. If promoting self-renewal, the gene related to self-renewal should be down-regulated upon KL treatment or AFF1 knock-down.

We appreciate this suggestion. We have extended our investigation to look at the genes related to self-renewal, including MYC and DNMT1. These two genes were not significantly altered with 1 or 3-hour KL treatment as seen with both mRNA and nascent RNA-seq data; however, both began to show downregulation with 24 hour KL treatment (new data included as Supplementary Figure 5m). In addition, these genes were downregulated with AFF1 and AFF4 knockdown RNA-seq. We have included western blot data with verification that DNMT1 and MYC are indeed also downregulated at the protein level with both KL treatment and knockdown (new data included as Supplementary Figure 2a,b, Supplementary Figure 3c-f). Interestingly, our AFF1 ChIP-seq data indicate that AFF1 binding is enriched near the promoters of MYC and DNMT1 (new data included in Supplementary Figure 5l). These data indicate that the SEC is involved in sustaining the

expression of key genes involved in self-renewal. However, the downregulation upon SEC perturbation appears to be a lot slower than the rapid-response genes with upregulation. We have included this in our discussion.

2. Line 138. The gene groups of cluster II show the up-regulation upon differentiation, although their genome-wide pol II occupancy decreased. What do you think of the reduced pol II level on gene body regions inducing the up-regulation of those genes?

We apologize for the confusion. However, the Pol II level of Cluster II on gene body does increase in the differentiation state. We have adjusted the Y-axis to make the change more visible in Figure 1d. We have also included comparisons of the undifferentiated and differentiated state in Figure 1j.

3. Line 135~138, Figure 1.c, e. I think the peak pattern of the example does not match the metaplot.

The reviewer made a great point. Not all genes clustered by the algorithm show exact the same pattern. We have included a more representative example, GRHL3, in the figure (Figure 1e).

4. Line 143~145. As you used the pausing index to explain the mechanism for regulation of the genes, the high pausing index shows the correlation of high gene expression. In figure a, however, the gene expression on cluster I seems to show no gene expression despite the high pol II pausing at TSS in an undifferentiated state.

We apologize for the confusion. The heatmap shown in Figure 1a reflects relative expression changes between the undifferentiated and differentiated keratinocytes. These values don't reflect the absolute high versus low gene expression levels. When comparing absolute expression levels, Cluster I genes show higher overall expression than Cluster II genes (**new data included as Reviewer Figure 6**). Thus, Cluster I genes are expressed and have high pausing index in progenitors, but also show increased expression coinciding with pause release in the differentiated condition.

5. Figure 1.i. P-TEFb phosphorylates DSIF and NELF to release the pol II pause. It means that the pause release and gene activation without P-TEFb is not persuasive. According to Liang et al (2018, Cell), the KL treatment causes degradation of AFF1/4 and the decrease of pol II elongation rate. However, 3-hour KL treatment makes several genes upregulated. How can you explain these genes upregulated?

We appreciate these questions from the reviewer. CDK9 inhibitors alone also did not mimic KL in inducing upregulation of these genes which was indicative that KL was not functioning by broadly inhibiting CDK9 (Figure 4a). To establish that gene activation after SEC perturbation still required P-TEFb, we treated cells with KL in combination with CDK9 inhibitors. The upregulation of differentiation activating genes with SEC perturbation was abolished with the addition of CDK9 inhibitors indicating that after KL treatment CDK9 was in an active state (Figure 4b). These findings gave us the first clue that a CDK9 activity switch, from the inactive state to the active state, could be involved with SEC perturbation. To further investigate this, we performed a number of experiments. First, we disrupted the inactive state of CDK9 by knocking down HEXIM1. Consistent with our observations from AFF1 knockdown and KL treatment, HEXIM1 knockdown upregulated differentiation, supporting that the inactive state of CDK9 is associated with suppression of differentiation gene expression in progenitor maintenance (Figure 4j-l). Using co-immunoprecipitation, we have now clarified that the interaction between AFF1 and HEXIM1 occurs in the progenitor-state keratinocytes, but not in the differentiated state; The AFF1 CDK9 interaction retains in both states (**new data included as Figure 4m,n**). These data also support that the CDK9 activity change, demonstrated by dissociation from HEXIM1, is involved in the differentiation process. Furthermore, using nascent RNA-seq, we examined gene expression at 1 hour of KL treatment to evaluate the earliest transcriptional response. In agreement with our previous observations, the majority of changes at the nascent RNA level are associated with gene upregulation (**new data included as supplementary Figure 5i**). Therefore, the current findings indicate that CDK9's transition from inactive to active state is involved in the upregulation of these genes.

We also compared our results to those published in Liang et al. 2018, which used the 293T cell line. Notably, a subset of genes is also upregulated in their RNA-seq data with 6 hours of KL treatment, including ATF3 which we identified as a key upregulated gene in keratinocytes (**new data included as Reviewer Figure 7a-b**). Their ChIP-seq data also showed AFF1 enrichment at ATF3 promoter (**new data included as Reviewer Figure 7c**). These data indicate that the SEC also have repressive roles for a subset of genes in other cell types, such as 293T cells. Also similar to their study, we observed reduction of both AFF1 and AFF4 at the protein level with 24 hours of KL treatment (**new data included as Supplementary Figure 2e-h**).

Lastly, we performed Pol II ChIP-seq after 1-hour and 3-hour KL treatment to determine if there are changes to Pol II pausing. We found only minor changes in Pol II pausing at upregulated, rapid-response genes after KL treatment. In the example of ATF3, Pol II binding was slightly increased in the gene body (new data included as Supplementary Figure 8a-g). Thus, KL treatment in this context appeared to de-stabilize the paused Pol II, rather than enhancing Pol II pausing.

To determine how pausing could influence these genes more directly, we looked at NELF. We knocked down NELFC/D and found only 10% of rapid-response genes and 10% of 24hr-KL differentially expressed genes to be also differentially expressed with NELF knockdown (new data included as Reviewer Figure 8a). Interestingly, however, we performed a preliminary ChIP-seq experiment to examine NELFA binding in progenitors and found 77% of upregulated, rapid-response genes were NELF direct targets (new data included as Reviewer Figure 8b-e). These data suggest that it is possible for these genes to be pause regulated, but multiple factors and mechanisms may be underlying control over pause release such that SEC inhibition alone is not sufficient for sustained pause release.

6. Figure 1.k, l, m. In figure 2, the RNA-seq DEG heatmaps upon 24-hour KL treatment were shown. I'm wondering their gene groups show similar patterns with those upon 3-hour treatment. It will be good if those data upon 24-hour KL are also shown.

We have included the percent of 24 hour KL-regulated genes that fall within each cluster of Pol II occupancy. While the vast majority of 3hr KL-regulated genes fell within "cluster I", 24-hour gene targets are more evenly spread across the three clusters (new data included as Figure 2j-l).

7. Line 203, supplementary figure 2a, b. The western blot band is not clear. I suggest the western data should be more obvious, like figure 4c.

We have now generated a higher-quality western blot showing AFF1 knockdown efficiency (new data included as Figure 3a). Sorry that it took us much longer to reach a higher quality, as the AFF1 antibody we used is not as robust as other antibodies used in this study for western.

8. Supplementary figure 2d. Figure 2d is written two times. AFF1sh-1/2 might be figure 2c.

We thank the reviewer for catching this! We have corrected the labeling in Supplementary figure 2.

9. Line 214~215. Figure 3c. What do you think of the reason for the accumulation of the regenerated red tissue? This difference between AFF1KD and AFF4KD is surprising, so I'm curious about the reason. In addition, AFF1sh-1/2 all shows this phenomenon?

We appreciate this question from the reviewer, and we found the findings to be reproducible between the two shRNAs (new data included as Supplementary Figure 3g). As supported by our RNA-seq data, AFF4 KD primarily impaired proliferation, whereas AFF1 KD upregulates differentiation-activating genes. In epidermis, the upper layers are the differentiated layers. When AFF1 KD were mixed together with control cells to seed on dermis, the regeneration of epidermal tissue involves a self-organization process. Cells with higher levels of differentiation, such as in the case of AFF1 but not AFF4 KD, were likely to have an advantage to localize in the upper more differentiation layer of epidermis.

10. Line 237~240. Figure 3k~m. This sentence is ambiguous to understand. The reason that AFF1 knock-down and KL treatment would show 'minimal' effect should be explained more. KL treatment causes the dissociation of P-TEFb, so AFF1 knock-down of the KL treatment sample will not give more changes than only the KL treatment sample. If that is the intention, I recommend the summation of the figure (k and l) and (k and m).

We have clarified this sentence in the text. If KL is acting on AFF1 to cause induction of differentiation, then we expected to see minimal changes in gene expression when AFF1 knockdown cells are treated with KL. While there is moderate additional upregulation of OVOL1, there is overall little additional upregulation of differentiation activators. We have summated k-m and included it in Supplementary Figure 3n,o.

11. Figure 3k~m. Different from the previous figure 2f~i, which type of KL was used?

Figure 3k~m was originally made to be relative to the DMSO treatment from each condition: non-targeting control, AFF1 sh1, or AFF1 sh2. In response to comment 10, we have summated these panels and made them relative to the non-targeting control DMSO condition (Supplementary Figure 3n,o). The scale in these

summed figures is now comparable to that seen in Figure 3l where AFF1 knockdown alone is shown. KL2 was used for these experiments.

12. Figure 4g-i. Why is only KL2 used in this RT-PCR?

Considering the consistency between KL1 and KL2 modulating gene expression as we observed from our RNA-seq, we used KL2 as a representative KL for this RT-PCR but included multiple shRNA's. The same approach was also taken by Zheng et al. (Genes & Development, 2021) with one representative KL used for the follow-up experiments.

13. Line 265~266. Figure 4g-i. It is not clear. Through figure 2, KL treatment itself seems to induce differentiation. Why do you mention 'unable to further induce differentiation?' In this sentence, what is the control, either control KD or HEXIM1KD+DMSO? As for question 10, I recommend the summation of the figure (g and h) and (g and i). Which explanation is plausible, HEXIM1 is also dissociated from P-TEFb, or P-TEFb can be activated despite the binding of HEXIM1 without AFF1? In figure 4a-b, P-TEFb can be activated after the dissociation from AFF1. If the control and KL2 treatment in the HEXIM1KD does not show differences, it seems that HEXIM1 itself cannot block the activation of P-TEFb.

We appreciate the reviewer pointing out the lack of clarity in the representation of these data. We have clarified this in the text and summed the data in Supplementary Figure 3n,o as well as in Supplementary Figure 4g,h. We have made these data relative to non-targeting control treated with DMSO. We propose that if KL is acting upon AFF1-HEXIM1, then in the context of HEXIM1 knockdown when HEXIM1 is absent, there will be minimal additional changes to gene expression. Minimal changes are seen in GRHL3 and ZNF750 expression when HEXIM1 knockdown cells are treated with KL; however, larger changes are seen with OVOL1 and PRDM1. Thus, it does appear that HEXIM1 itself is not entirely sufficient to block P-TEFb at a subset of genes.

Given the body of literature suggesting that while associated with HEXIM1, P-TEFb is in an inactive state, the most plausible explanation is that HEXIM1 is also dissociated from P-TEFb after KL treatment. Indeed, through co-immunoprecipitation experiments, we demonstrated that KL causes dissociation of CDK9 from HEXIM1 (new data included as Figure 8a,b). It is therefore likely that KL is causing induction of differentiation by breaking apart CDK9 from both AFF1 and HEXIM1.

14. Figure 5e. I could understand the reason that RNA-seq upon 3hr KL treatment. But I wonder how many overlapping genes exist upon 24hr KL treatment.

We overlapped HEXIM1 and AFF1 ChIP-seq data with genes differentially expressed after 24 hours of KL treatment. 271 genes differentially expressed after 24-hour KL treatment were identified as direct targets (new data included as Supplementary Figure 5d).

15. Figure 5i-k. The ChIP-seq example on the region of the differentiation-activating TFs (GRHL3, OVOL1, PRDM1, ZNF750) is required.

We have added ChIP-seq browser tracks at differentiation activators (new data included as Supplementary Figure 5l). These genes show little to no enrichment of AFF1 or HEXIM1. This is consistent with the model that AFF1 and HEXIM1 directly regulate rapid-response genes setting off a cascade of differentiation.

16. Figure 5i-k. How is the distribution of RNAP II upon KL treatment, AFF1KD, or HEXIM1KD?

We performed Pol II ChIP-seq with KL treatment as well as with AFF1 or HEXIM1 knockdown. In keratinocytes treated for 1 hour or 3 hours with KL, the changes in pausing-index distribution at neither time point was statistically significant; However, differential analysis using EdgeR was able to detect subtle changes in pausing indices that varied by time point at different genes (new data included as Supplementary Figure 8a-c). In the example of ATF3, increased Pol II binding along the gene body was visible with 1-hr KL treatment (new data included as Supplementary Figure 8g). These data suggest that KL treatment transiently destabilized Pol II from the paused state, but this was insufficient to fully convert Pol II towards stable elongation. It is likely that the contributions from other elongation regulators are needed to stabilize Pol II in the elongation state in fully differentiated keratinocytes. We observe similar subtle changes to pausing with 30min and 1hr TPA treatments (new data included as Supplementary Figure 8d-f).

In keratinocytes with HEXIM1 knockdown, we observed significant changes in Pol II pausing distribution from both shRNAs. However the results from the two AFF1 shRNAs were more variable. Differential analyses

using EdgeR, however, were able to detect changes of pausing indices at different genes (new data included as Reviewer Figure 9). Overall, the knockdown of AFF1 or HEXIM1 was still not sufficient to fully convert Pol II from the paused to a stable elongation state as observed in fully differentiated keratinocytes, highlighting the participation from other regulators than the SEC in modulating Pol II dynamics.

17. To make figure 3~4 more reliable, P-TEFb ChIP-seq upon AFF1KD and HEXIM1KD is required. The P-TEFb distribution will not be changed? If not so, how could the dissociation of P-TEFb from SEC induce the gene activation?

We appreciate this suggestion. As AFF1 or HEXIM1 knockdown drastically impaired keratinocyte proliferation, ChIP-seq would only be feasible if the antibody were exceptional. We were able to perform CDK9 ChIP-seq with HA-tagged CDK9, but this experiment required at least 10 million cells to reach high-quality sequencing data. To circumvent this technical challenge, we performed HA-CDK9 ChIP-seq in keratinocytes treated KL or TPA and compared to DMSO control. Interestingly, we found that the distribution of CDK9 binding changed with KL or TPA treatment (new data included as Figure 8d-h). We calculated “traveling ratios” to determine the ratio of CDK9 at the gene body relative to the promoter. This ratio increased with drug treatment indicating CDK9 was being released along the gene bodies.

These data suggest CDK9 is being released from the inactive state with HEXIM1 to an active state upon treatment with KL or TPA. Active P-TEFb can associate with BRD4. We asked whether BRD4 could be involved in facilitating differentiation induction with KL. We treated keratinocytes with KL alone or with KL in combination with a BRD4 inhibitor (JQ1 or CPI203). Although BRD4 inhibition abolished the induction of differentiation-activating TFs, the induction of rapid-response SEC targets, ATF3 and DUSP1, was not affected (new data included as Supplementary Figure 8h). These data indicate that an alternative mechanism, other than BRD4, is involved in binding to the active CDK9 to facilitate the induction of the rapid-response genes. This is consistent with Zheng et al. (Genes & Development, 2021) showing that BRD4 was not necessary for rapid activation of gene expression under heat shock conditions.

18. Figure 6. In the title of this manuscript, SEC-pTEFb itself seems to induce the keratinocyte differentiation. However, the differentiation seems to result from an indirect effect through figure 6. In addition, the 3hr KL treatment is sufficient to induce the differentiation-activating TFs. Which genes are first activated after KL treatment, ATF3 or differentiation-activating TFs? Short treatment of KL should be required.

We appreciate this interesting question from the reviewer. Indeed, our data from Figure 6 suggest that the upregulation of ATF3 is sufficient to promote the induction of several differentiation-activating TFs. To further compare the timing of direct targets versus these differentiation-activating TFs, we performed nascent RNA-sequencing and added a shorter KL treatment of just 1 hour. Within 1 hour, direct targets ATF3, DUSP1, and RND3 were already significantly upregulated, prior to the upregulation of differentiation-activating TFs such as ZNF750 and OVOL1 (new data included as Figure 5g).

19. Line 330~332. Supplementary figure 5b. Although some articles suggest the calcium-mediated keratinocyte differentiation, no change of ATF3 gene in calcium insertion might implicate that ATF3 itself is not sufficient to induce the differentiation. What about the clonogenic assay upon calcium treatment?

We agree that elevated calcium can induce a subset of differentiation markers even at sub-confluent conditions. For example, Mahanty et al. (*Cell Death & Disease*, 2019) used this condition for their study. One key difference is timing. Our data shown in Supplementary Figure 7b were generated with only 3-hours of calcium treatment in sub-confluent conditions, while the condition used by Mahanty et al. was 60 hours. In our standard calcium-induced differentiation protocol, we add calcium when keratinocytes reach full confluence for four days. Under these conditions, we detected robust ATF3 expression (new data included as Supplementary Figure 7a). We apologize for this confusion, and we have further clarified this in the result section.

20. Line 341~345. Figure 7a. If ATF3 directly induces the differentiation activating genes, ATF3 expression should be maintained while those genes are activated. Why does the expression of the ATF3 gene decrease after 3hr?

We thank the reviewer for this great question! ATF3 mRNA expression is upregulated at both the 1-hour and 3-hour time points, although the level is higher at 1 hour. To determine the protein expression of ATF3, we performed western blots. We found that the protein level of ATF3 is elevated after 1 hour and maintained at the

3-hour time point (new data included as Supplementary Figure 7c,d), indicating that the upregulation of ATF3 protein is maintained at least at the 3-hour time point.

21. Figure 7c, d. What about the summation of the figure 7c and 7d? The graph of Figure 7d is too short of comparing.

We agree with the reviewer that the summation of these figures can provide more information. The summation data are now included as Supplementary Figure 7f. In addition to TPA treatment in combination with KL, we also performed TPA treatment on cells with AFF1 or HEXIM1 knockdown. As seen with KL treatment, after AFF1 or HEXIM1 knockdown, TPA caused minimal changes to gene expression (new data included as Supplementary Figure 7g-j).

22. Figure 7e. What if Cdk9 inhibition after KL, siAFF1, or siHEXIM1?

We agree with the reviewer that it is important to compare CDK9 inhibition after KL and AFF1 knockdown or HEXIM1 knockdown. We previously included CDK9 inhibition after KL as Figure 4b. We have now additionally performed CDK9 inhibition with AFF1 and HEXIM1 knockdown, and found that CDK9i drastically blocked the activation of differentiation markers in both cases (new data included as Supplementary Figure 4a,b,e,f).

23. Line 429 ~ 430. It might be possible that the difference with the previous data related to pol II pausing might result from the use of primary cell. How about showing the difference between primary cell and immortalized cell upon KL or AFF1KD treatment?

It is a good question! We have included the analysis in a reviewer figure (new data included as Reviewer Figure R7), and we have updated the contents in discussion. We analyzed 293T cell 6-hour KL data from Liang et. al. 2018 and consistent with what they found, identified a subset of upregulated genes. Among these genes was ATF3 which was also upregulated in our system; however, the low percent overlap between keratinocyte and 293T cell upregulated genes suggests that the SEC may have cell-type specific functions. The activating functions of the SEC observed in 293T cells are also consistent in keratinocytes. We saw that disruption of the SEC by knockdown or inhibitor treatment led to loss of expression of cell cycle activating genes (new data included as Supplementary Figure 2a,b, Supplementary Figure 3e,f). Notably, a central SEC target, MYC, identified in Liang et al. 2018 was also downregulated in our study at the protein level.

24. Line 562. The exact number of cycles might be written.

We have added the number of cycles required for sonication to the methods section.

Reviewer #3:

In this study, Llyod et al. carried out Pol II ChIP-seq and combined this analysis with their previously reported RNA-seq data to determine Pol II dynamics during KC differentiation. This analysis allowed them to identify a cluster of genes that were highly enriched for paused Pol II in undifferentiated KCs and robust Pol II elongation in differentiated KCs. Interestingly, disruption of PTEFb-SEC association for 3 hours results in upregulation of several Cluster I genes in UD KCs, and disruption for 24 hours leads to a larger gene expression change which includes upregulation of several genes that are expressed in DF KCs. Further studies revealed that loss of AFF1, and not AFF4, results in the upregulation of genes associated with epidermal differentiation. Additionally, HEXIM1 synergistically associates with AAF1-SEC to maintain target gene repression. The authors further show that the activity switch of SEC-PTEFb mediates the early events of PKC signaling which is important for keratinocyte differentiation.

Overall, this study has identified that the activity switch of SEC-PTEFb is responsible for regulating the earliest events that promote the fate-choice decisions between self-renewal and differentiation. Findings reported in this paper add to the knowledge of Pol II dynamics between UD and DF states of KC and will be of interest to researchers working in the broader community of gene regulation during differentiation. Although this paper does a thorough job in addressing the gene expression changes under the control of HEXIM-SEC-PTEFb mediated RNA Pol II pausing in maintaining progenitor fate, it fails to address some fundamental questions mentioned below. The following comments and concerns need to be addressed in this manuscript before it is considered for publication.

Major Comments:

1. In the introduction the authors mention that while the inactive form of PTEFb associates with a complex containing HEXIM1, the active form of PTEFb associates with SEC. But in this study, they conclude that HEXIM1 works synergistically with AFF1-SEC by maintaining CDK9 in an inactive state. Therefore, to fully support the model reported in Figure 4m, the authors need to conduct biochemical assays to show the physical interaction of these proteins in KCs. An IP with antibodies against CDK9 and pCDK9 and then followed by western blot on IP product to show an association of HEXIM1 and AFF1 with inactive pTEFb is absolutely needed. Moreover, is 7SK snRNA also part of this synergistic complex? Depletion of 7SK snRNA results in untethering of HEXIM to inactive PTEFb which allows for PTEFb phosphorylation and Pol II elongation. Please address. Add reference to the new paper AFF interaction with pCDK9.

We thank the reviewer for suggesting biochemical assays to strengthen our results. All the antibodies recognizing AFF1, CDK9/pCDK9, and HEXIM1 we validated are rabbit polyclonal antibodies. To overcome the technical challenge of performing co-immunoprecipitation, we expressed HA-AFF1 in keratinocytes and performed immunoprecipitation using a monoclonal, mouse HA antibody. We found that AFF1 and HEXIM1 interact in the progenitor-state keratinocytes but not in the differentiation-state keratinocytes. In contrast, the association between AFF1 and CDK9/pCDK9 persists in differentiation (new data included as Figure 4m,n).

We have incorporated the potential role for 7SK in working with the AFF1-HEXIM1 repressive complex in the discussion. A recently published paper (Bandiera et. al. 2021) reported that 7SK knockdown in human keratinocytes induced upregulation of differentiation markers such as TGM and INV, which is consistent with our observed differentiation induction from HEXIM1 knockdown.

2. The authors show that HEXIM1 and AFF1 specifically target “rapid response” genes in UD KCs to stall Pol II at the proximal promoter regions based on ChIP-seq results. However, the authors do not address what is unique about these genes that result in the association of HEXIM1-inactive pTEFb with AFF1-containing SEC. Are there DNA sequences common between these differentiation-promoting genes that allow for this unique targeting in UD KCs? Are there epigenetic signatures unique to these genes?

We appreciate this question from the reviewer. We have performed a number of motif searches using different parameters, trying to identify potential motifs associated with the rapid response genes. However, there does not appear to be a single motif that is represented in the majority of these rapid-response genes.

Leveraging the publicly available ENCODE ChIP-seq data of various histone modifications/modifiers in keratinocytes (NHEK), we compared their relative enrichment in the AFF1 ChIP-seq peak regions associated with the “rapid response genes” versus the genes changed with 24-hours of KL treatment. In summary, the “rapid-response gene”-associated AFF1 peaks are characterized by much lower levels of EZH2 binding and H3K27me3, but higher H4K20me1 enrichment (new data included as Reviewer Figure 10).

3. Does PKC activation result in dissociation of HEXIM1 from these repressed genes allowing for gene expression during differentiation? To show a direct correlation of AFF1-PTEFb-HEXIM1 disruption upon PKC signaling, and Pol II elongation resulting in expression of differentiation promoting TFs, the authors need to conduct ChIP analysis of AFF1, HEXIM1, and Pol II on target genes in UD KCs treated with TPA. Also, an IP of CDK9 in KCs treated with TPA can also clarify if the AFF1-PTEFb-HEXIM1 is altered. Also, does PKC simply result in dissociation of HEXIM1, or its expression is diminished upon PKC signaling?

We appreciate these great questions! To better understand how TPA influences the action of HEXIM1 and pTEFb, we performed co-immunoprecipitation in keratinocytes expressing HA-CDK9. We found that the interaction between HEXIM1 and CDK9 was lost with 3-hour KL treatment or with 1-hour TPA treatment (new data included as Figure 8a-c). These data suggest that KL or TPA treatment was sufficient to break apart CDK9 from the inactive HEXIM1 complex, allowing for rapid activation of the direct target genes.

We assessed expression of HEXIM1 at the protein level after activation of PKC signaling with 1 or 3 hours of TPA treatment. We detected no significant changes in HEXIM1 expression (new data included as Reviewer Figure 11a,b).

Interestingly, Pol II ChIP-seq on keratinocytes treated with TPA for 30min or 1 hour revealed only minor changes to Pol II pausing (new data included as Supplementary Figure 7d-f), in contrast to the drastic pause release as observed in terminal differentiation. These data suggest that TPA treatment destabilized the paused Pol II, yet additional mechanisms may be required to stabilize Pol II in the elongation state. We also performed AFF1 and HEXIM1 ChIP-qPCR with TPA and KL treatment (new data included as Supplementary Figure 8i,j).

We found a modest reduction in AFF1 binding but no significant changes to HEXIM1 binding with TPA or KL treatment. Thus, the changes in CDK9 activity may not be a result of reduced AFF1 and HEXIM1 chromatin binding, but rather a result of CDK9 being released from HEXIM1 as indicated by our co-immunoprecipitation results (new data included as Figure 8a-c).

Other Comments:

1. Given that the authors see a clear distinction in Pol II occupancy at TSS and gene body in UD vs DF states of KC, how does the distribution of pSer5 vs pSer2 look like in these states? Pol II CTD accumulates pSer2 as it progresses through a gene body, therefore is this accumulation evident in the DF state of KC? Can the authors perform IF or Western Blot analysis with pSer2 and pSer5 in UD and DF to show if the phosphorylated state of Pol CTD residues mirrors the ChIP data? Alternatively, ChIP-qPCR using pSer2 and pSer5 antibodies on select Cluster I genes can also address this.

We assessed the levels of phosphorylated Pol II during keratinocyte differentiation by western blotting. Both Ser2 and Ser5 phosphorylation are decreased in the course of keratinocyte differentiation (new data included as Reviewer Figure 12a-c).

To investigate the dynamics of PolIII-Ser2P and PolIII-Ser5P at specific subsets of genes, we performed ChIP-seq with PolIII-Ser2P and PolIII-Ser5P antibodies. At genes downregulated during differentiation or with 24-hour KL treatment, total Pol II-Ser2P and total PolIII-Ser5P decreased in occupancy. At genes upregulated during differentiation or with 24-hour KL treatment, Pol II-Ser2P increased occupancy along gene bodies and at the 3' end (new data included as Reviewer Figure 12d-k). At these upregulated genes, PolIII-Ser5P also increased in occupancy, but was highest in promoter regions. Rapid-response genes generally followed trends seen in upregulated 24-hour KL and differentiation genes (new data included as Reviewer Figure 12l)

2. It looks like Cluster II genes have lower Pol II occupancy at TSS even in UD KC but accumulate Pol II elongation in DF KCs. Is this because chromatin dynamics (repressed chromatin domains) play a role in inhibiting Pol II access to Cluster II genes at UD state? Moreover, Cluster III genes are downregulated in DF KCs, indicating that they are actively transcribed in UD KCs. Therefore, it is important to include Pol II enrichment in gene bodies of Cluster III genes in Sup Fig 1e – which should be high in UD and low in DF compared to Cluster I.

Cluster II genes indeed have lower Pol II occupancy at the TSS in both UD and DF states, as compared to Cluster I. Leveraging the H3K27me3 ChIP-seq data generated in keratinocytes from ENCODE, we compared the level of H3K27me3 in Cluster I and Cluster II. 13.5% of Cluster II genes have H3K27me3 ChIP-seq peaks at the TSS, while only 1.2% of cluster I genes have H3K27me3 peaks at their TSSs. Thus, these findings are in agreement with the prediction from the reviewer that Cluster II genes are more associated with repressed chromatin domains.

We have also added the gene body Pol II enrichment for Cluster III genes. There is a decrease in total Pol II in the DF condition (new data included as Figure 1j).

3. The authors mention the number of shared genes between KL treated and DF KCs, but it will be helpful if they can also indicate how many of the shared genes that are upregulated in KL treated UD KCs are also expressed in DF KCs? Is there a statistical significance in this overlap? Maybe a GSEA analysis using DF gene set as the background will be helpful.

We defined genes expressed in DF KCs with a cutoff of > 5 average counts in RNA-seq across three replicates. 950 of 972 KL-upregulated genes were also expressed in DF KC. This overlap is statistically significant ($P < 0.001$, Fisher's exact test) (new data included as Reviewer Figure 13). We attempted GSEA with KL 24-hour RNA-seq data compared to DF, and results were not clearly conclusive.

4. The authors at the end of Figure 2 conclude that "SEC-PTEFb disruption progressively upregulates terminal differentiation and diminishes proliferation", while the latter was shown by IF of Ki67 and cell numbers, the authors need to carry out IF of classical differentiation markers to show that UD cells when treated with KL take up on a differentiated fate.

We appreciate this suggestion and have included p21 staining in the figures (new data included as Supplementary Figure 2c,d). The initial keratinocyte differentiation commitment involves the upregulation of p21 (Freije et al., 2012), and we also observe that p21 is upregulated with KL treatment at the 24 hour time point.

5. The authors carried out PCA analysis to show KL treated UD KCs were similar to that of AFF1sh KCs, but they don't report the number of shared genes that are upregulated in both instances. The authors need to provide a Venn or a graph showing how many genes upregulated upon AFF1sh are also upregulated in 24-hour KL treatment.

We have included a pie chart showing the percent of AFF1 upregulated genes also upregulated in 24-hour KL treatment. 32% of AFF1 upregulated genes are significantly upregulated after 24-hour KL treatment. We also generated a heatmap of all AFF1 upregulated genes (new data included as Supplementary Figure 3l,m). The majority of these genes show a similar trend in gene expression changes with KL treatment. It is likely some of the differences seen can be attributed to differences in timing between knockdown (~4 days) and drug treatment (24 hours).

6. Fig. 3k-m is a bit confusing. AFF1 KD leads to significant upregulation of TFs, as shown in Fig. 2h, but in 3l and m in the DMSO panel, the upregulation of these TFs is not as apparent (y-scale on 2h was up to 20-30 while on 3l is below 5). Please address this discrepancy. Also, why would AFF1 KD mediated upregulation "mitigate" the effects of KL treatment, if they are both targeting the SEC-PTEFb association? I have the same concern with Fig 4f and 4h. Why doesn't the HEXIM KD without KL show the same level of upregulation as compared to shown in 4f? Please explain.

We apologize for the confusion. In the current Figure 3q-s, we re-organized the data by comparing KL treatment versus DMSO within the same group of keratinocytes treated by AFF1 knockdown or non-targeting control. Similarly in figure 4g-l, each panel is a comparison between KL treatment versus DMSO control within the same group of keratinocytes, with HEXIM1 knockdown or non-targeting control. To make these data clearer we have summated these data in the supplemental figures (Supplementary Figure 3n,o, and Supplementary Figure 4g,h), where everything is shown relative to a non-targeting, DMSO control. The y axis for these data are now comparable to data showing knockdown alone. These experiments were designed to determine if KL induced differentiation is dependent on the intact function of AFF1 for example. With AFF1 knockdown, there was little further upregulation of differentiation induced by KL. We have clarified these experiments in the results section.

Reviewer Figure 1. ATF3 Overexpression (OE) upregulates PRDM1. **(a,b)** Western blot and quantification of ATF3 OE probed with PRDM1 (n = 3 technical replicates, **P < 0.05, Unpaired t-test).

Reviewer Figure 2. AFF1/4 knockdown impairs epidermal self-renewal. **(a,b)** Quantification of epidermal thickness in competition assays (n = 3 biological replicates) **(c)** *In vivo* competition assays showing reduced AFF1 or AFF4 knockdown cells in epidermal regeneration, as compared with non-targeting control (scale bar = 125 μM). **(d,e)** Keratin 10 staining and quantification of fluorescent intensity in regenerated epidermis with non-targeting control, AFF1, or AFF4 knockdown (scale = 125 μm , n=7 technical replicates).

Reviewer Figure 3. Calcium treatment does not affect upregulation of differentiation induced by AFF1 knockdown. **(a,b)** qRT-PCR of AFF1 knockdown cells treated with Calcium relative to non-targeting control.

Reviewer Figure 4. (a) qRT-PCR showing upregulation of differentiation activators with overexpression of mouse ATF3 in human keratinocytes **(b)** 3-hour KL treatment RNA-seq compared to 3-hour TPA treatment RNA-seq.

Reviewer Figure 5. SEC inhibition with KL does not significantly alter global RNA production. (a) Representative images of nascent RNA imaging of keratinocytes treated with SEC inhibitor KL2 or DMSO control for 24 hours. This technique leverages ethylene uridine (EU) ribonucleotide homolog containing an alkyne reactive group, and the detection of nascent RNA utilizes “click” reaction to incorporate azide-containing Alexa fluor 594. DNA is visualized by Hoechst 33342. **(b)** Quantification of the average signal across 3 replications (n=12 images/replicate, P=0.97, unpaired t-test, scale bar=50uM).

Reviewer Figure 6. Total Expression level of genes differentially expressed in differentiation. **(a-c)** Heatmaps showing Log₂ of normalized counts generated from undifferentiated and differentiated keratinocyte RNA-seq data at clusters identified in Fig. 1a.

Reviewer Figure 7. Comparison of KL-treated 293T cells to human keratinocytes. **(a)** Overlap of RNA-seq from 6-hour KL treated 293T cells (Liang et. al. 2018) with RNA-seq data from human keratinocytes treated with KL for 3 hours. **(b)** Heatmap showing gene expression changes from 6hr KL-treated 293T cells and 3-hr KL-treated keratinocytes at 92 direct, rapid-response genes. **(c)** Genome browser tracks showing AFF1 and Pol II enrichment in 293T cells and keratinocytes (KC) at ATF3.

Reviewer Figure 8. NELF Knockdown and ChIP-seq compared to KL-regulated genes **(a)** Venn diagram comparing genes differentially expressed with NELFC/D knockdown to direct, rapid-response genes and 24-hour KL differentially expressed genes. **(b)** Venn diagram comparing NELFA ChIP-seq direct target genes to direct, rapid-response genes. **(c-e)** Genome browser tracks showing NELFA ChIP-seq enrichment at ATF3, DUSP1, and RND3.

Reviewer Figure 9. Pol II Pausing Changes with AFF1 or HEXIM1 Knockdown (KD) at direct, rapid-response genes **(a,b)** Pol II Pausing Index distribution with AFF1 or HEXIM1 knockdown at 92 rapid-response genes (**** $P < 0.001$, *** $P < 0.01$, * $P < 0.1$, ns = not significant, unpaired t-test). **(c)** Heatmap showing dynamic pausing indices at 92 rapid-response genes with AFF1 or HEXIM1 knockdown.

Reviewer Figure 10. Chromatin modifications and regulators associated with rapid vs. late responding KL target genes. Bar graph showing the percentage of AFF1 peaks overlap with the peaks of other ChIP-seq data, comparing the “direct, rapid response genes (Direct, Rapid)” with the “upregulated genes with 24-hour KL treatment (KL 24 UP)” and the “downregulated genes with 24-hour KL treatment (KL 24 Down)”.

Reviewer Figure 11. HEXIM1 protein levels do not significantly change with TPA treatment. **(a,b)** Western blot and quantification of HEXIM1 protein level with 1 or 3-hour TPA treatment (n=3, ns P > 0.1, unpaired t-test).

Reviewer Figure 12. Pan-Pol II, Pol II Serine 2 (PolII-Ser2P) and Serine 5 (PolII-Ser5P) phosphorylation dynamics in keratinocyte differentiation. **(a-c)** Western blot and quantification of Pan-Pol II, PolII-Ser2P and PolII-Ser5P protein level in progenitors (UD), and at 48hr (D2) and 96hr (D4) of differentiation induced by 1.2mM CaCl₂ and confluency. **(d-k)** PolII-Ser2P and PolII-Ser5P average profile plots at the genes up or downregulated in epidermal differentiation or in 24-hr KL treatment. **(l)** Genome browser tracks showing PolII-Ser2P and PolII-Ser5P enrichment at ATF3, DUSP1, and RND3 in undifferentiated (UD) and differentiated (DF) keratinocytes.

Reviewer Figure 13. KL-Upregulated Genes are expressed in epidermal differentiation. Venn diagram showing overlap between genes expressed in differentiation with genes upregulated with 24-hour KL treatment.

REVIEWERS' COMMENTS

Reviewer #1 (Remarks to the Author):

Congratulations on the successful revision of the manuscript and the great work. I look forward to have the paper accessible to the scientific community. Well done.

Reviewer #2 (Remarks to the Author):

I am satisfied with the changes they brought to the newly revised version, and recommend publication in Nature Communications.

Reviewer #3 (Remarks to the Author):

Llyod et. al., have thoroughly addressed my questions and concerns and I, therefore, recommend the manuscript for publication. However, there are some very minor comments listed below, when addressed, will improve the overall presentation of this extensive work.

Minor Comments:

1. Several paragraphs in the introduction end with "incompletely understood". It is a little repetitive and I suggest using synonyms to improve the comprehension.
2. Figure 4g-i and 7c: Significance is not indicated on the graphs like it has been shown in other qRT-PCR data.

Reviewer #1:

Congratulations on the successful revision of the manuscript and the great work. I look forward to have the paper accessible to the scientific community. Well done.

We thank Reviewer #1 for all the suggestions, which helped to strengthen this paper in the revision process.

Reviewer #2:

I am satisfied with the changes they brought to the newly revised version, and recommend publication in Nature Communications.

We appreciate the all questions and suggestions from Reviewer #2 in the revision process, which were very helpful for us to clarify the molecular mechanisms.

Reviewer #3:

Llyod et. al., have thoroughly addressed my questions and concerns and I, therefore, recommend the manuscript for publication. However, there are some very minor comments listed below, when addressed, will improve the overall presentation of this extensive work.

We thank reviewer #3 for all the helpful questions and comments in the revision process!

Minor Comments:

1. Several paragraphs in the introduction end with “incompletely understood”. It is a little repetitive and I suggest using synonyms to improve the comprehension.

We appreciate this comment, and we have rephrased these sentences in the introduction to decrease the repetitiveness.

2. Figure 4g-i and 7c: Significance is not indicated on the graphs like it has been shown in other qRT-PCR data.

We thank the reviewer for pointing this out. Because qRT-PCR data are shown as technical rather than biological replicates, we have removed statistical testing showing p values on all qRT-PCR panels. This has also improved consistency as now all data from these experiments are shown in the same way.